# Contrastive Neural Ratio Estimation

**Benjamin Kurt Miller**
University of Amsterdam
b.k.miller@uva.nl

**Christoph Weniger**
University of Amsterdam
c.weniger@uva.nl

**Patrick Forré**
University of Amsterdam
p.d.forre@uva.nl

## Abstract

Likelihood-to-evidence ratio estimation is usually cast as either a binary (NRE-A) or a multiclass (NRE-B) classification task. In contrast to the binary classification framework, the current formulation of the multiclass version has an intrinsic and unknown bias term, making otherwise informative diagnostics unreliable. We propose a multiclass framework free from the bias inherent to NRE-B at optimum, leaving us in the position to run diagnostics that practitioners depend on. It also recovers NRE-A in one corner case and NRE-B in the limiting case. For fair comparison, we benchmark the behavior of all algorithms in both familiar and novel training regimes: when jointly drawn data is unlimited, when data is fixed but prior draws are unlimited, and in the commonplace fixed data and parameters setting. Our investigations reveal that the highest performing models are distant from the competitors (NRE-A, NRE-B) in hyperparameter space. We make a recommendation for hyperparameters distinct from the previous models. We suggest a bound on the mutual information as a performance metric for simulation-based inference methods, without the need for posterior samples, and provide experimental results.

## 1 Introduction

We begin with a motivating example: Consider the task of inferring the mass of an exoplanet $\boldsymbol{\theta}_o$ from the light curve observations $\boldsymbol{x}_o$ of a distant star. We design a computer program that maps hypothetical mass $\boldsymbol{\theta}$ to a simulated light curve $\boldsymbol{x}$ using relevant physical theory. Our simulator computes $\boldsymbol{x}$ from $\boldsymbol{\theta}$, but the inverse mapping is unspecified and likely intractable. *Simulation-based inference* (SBI) puts this problem in a probabilistic context [13, 64]. Although we cannot analytically evaluate it, we assume that the simulator is sampling from the conditional probability distribution $p(\boldsymbol{x} \,|\, \boldsymbol{\theta})$. After specifying a prior $p(\boldsymbol{\theta})$, the inverse amounts to estimating the posterior $p(\boldsymbol{\theta} \,|\, \boldsymbol{x}_o)$. This problem setting occurs across scientific domains [1, 7, 10, 11, 29] where $\boldsymbol{\theta}$ generally represents input parameters of the simulator and $\boldsymbol{x}$ the simulated output observation. Our design goal is to produce a surrogate model $\hat{p}(\boldsymbol{\theta} \,|\, \boldsymbol{x})$ approximating the posterior for any data $\boldsymbol{x}$ while limiting excessive simulation.

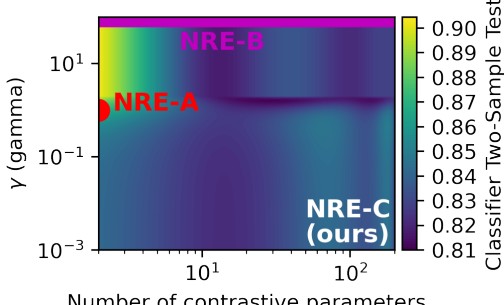

Figure 1: Conceptual, interpolated map from investigated hyperparameters of proposed algorithm NRE-C to a measurement of posterior exactness using the Classifier Two-Sample Test. Best 0.5, worst 1.0. Red dot indicates NRE-A's hyperparameters, $\gamma = 1$ and $K = 1$ [30]. Purple line implies NRE-B [16] with $\gamma = \infty$ and $K \geq 1$. NRE-C covers the entire plane, generalizing other methods. Best performance occurs with $K > 1$ and $\gamma \approx 1$, in contrast with the settings of existing algorithms.

Density estimation [5, 54, 55] can fit the likelihood [2, 15, 43, 56] or posterior [6, 21, 42, 53] directly; however, an appealing alternative for

36th Conference on Neural Information Processing Systems (NeurIPS 2022).

practitioners is estimating a *ratio* between distributions [12, 16, 30, 34, 69]. Specifically, the likelihood-to-evidence ratio $\frac{p(\boldsymbol{\theta}\,|\,\boldsymbol{x})}{p(\boldsymbol{\theta})} = \frac{p(\boldsymbol{x}\,|\,\boldsymbol{\theta})}{p(\boldsymbol{x})} = \frac{p(\boldsymbol{\theta},\boldsymbol{x})}{p(\boldsymbol{\theta})p(\boldsymbol{x})}$. Unlike the other methods, ratio estimation enables easy aggregation of independent and identically drawn data $\boldsymbol{x}$. Ratio and posterior estimation can compute bounds on the mutual information and an importance sampling diagnostic.

Estimating $\frac{p(\boldsymbol{x}\,|\,\boldsymbol{\theta})}{p(\boldsymbol{x})}$ can be formulated as a binary classification task [30], where the classifier $\sigma \circ f_{\boldsymbol{w}}(\boldsymbol{\theta},\boldsymbol{x})$ distinguishes between pairs $(\boldsymbol{\theta},\boldsymbol{x})$ sampled either from the joint distribution $p(\boldsymbol{\theta},\boldsymbol{x})$ or the product of its marginals $p(\boldsymbol{\theta})p(\boldsymbol{x})$. We call it NRE-A. The optimal classifier has

$$f_{\boldsymbol{w}}(\boldsymbol{\theta},\boldsymbol{x}) \approx \log\frac{p(\boldsymbol{\theta}\,|\,\boldsymbol{x})}{p(\boldsymbol{\theta})}. \qquad (1)$$

Here, $\sigma$ represents the sigmoid function, $\circ$ implies function composition, and $f_{\boldsymbol{w}}$ is a neural network with weights $\boldsymbol{w}$. As a part of an effort to unify different SBI methods and to improve simulation-efficiency, Durkan et al. [16] reformulated the classification task to identify which of $K$ possible $\boldsymbol{\theta}_k$ was responsible for simulating $\boldsymbol{x}$. We refer to it as NRE-B. At optimum

$$g_{\boldsymbol{w}}(\boldsymbol{\theta},\boldsymbol{x}) \approx \log\frac{p(\boldsymbol{\theta}\,|\,\boldsymbol{x})}{p(\boldsymbol{\theta})} + c_{\boldsymbol{w}}(\boldsymbol{x}), \qquad (2)$$

where an additional bias, $c_{\boldsymbol{w}}(\boldsymbol{x})$, appears. $g_{\boldsymbol{w}}$ represents another neural network. The $c_{\boldsymbol{w}}(\boldsymbol{x})$ term nullifies many of the advantages ratio estimation offers. $c_{\boldsymbol{w}}(\boldsymbol{x})$ can be arbitrarily pathological in $\boldsymbol{x}$, meaning that the normalizing constant can take on extreme values. This limits the applicability of verification tools like the importance sampling-based diagnostic in Section 2.2.

The $c_{\boldsymbol{w}}(\boldsymbol{x})$ term also arises in contrastive learning [23, 71] with Ma and Collins [45] attempting to estimate it in order to reduce its impact. We will propose a method that discourages this bias instead. Further discussion in Appendix D.

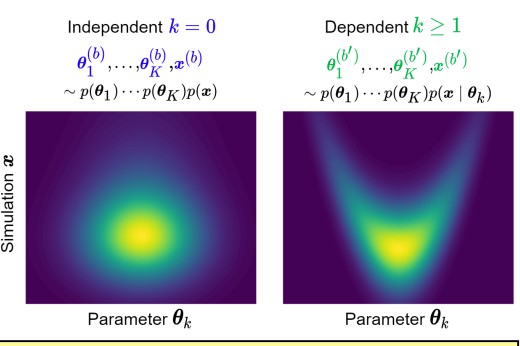

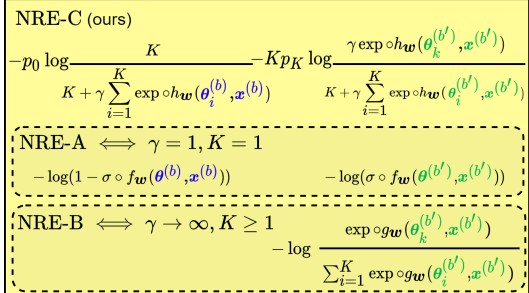

Figure 2: Schematic depicts how the loss is computed in NRE algorithms. $(\boldsymbol{\theta},\boldsymbol{x})$ pairs are sampled from distributions at the top of the figure, entering the loss functions as depicted. NRE-C controls the number of contrastive classes with $K$ and the weight of independent and dependent terms with $p_0$ and $p_K$. NRE-C generalizes other algorithms. Hyperparameters recovering NRE-A and NRE-B are listed next to the name within the dashed areas. Notation details are defined in Section 2.1.

There is a distinction in deep learning-based SBI between *amortized* and *sequential* algorithms which produce surrogate models that estimate any posterior $p(\boldsymbol{\theta}\,|\,\boldsymbol{x})$ or a specific posterior $p(\boldsymbol{\theta}\,|\,\boldsymbol{x}_o)$ respectively. Amortized algorithms sample parameters from the prior, while sequential algorithms use an alternative proposal distribution–increasing efficiency at the expense of flexibility. Amortization is usually necessary to compute diagnostics that do not require samples from $p(\boldsymbol{\theta}\,|\,\boldsymbol{x}_o)$ and amortized estimators are empirically more reliable [31]. Our study therefore focuses on amortized algorithms.

**Contribution**  We design a more general formulation of likelihood-to-evidence ratio estimation as a multiclass problem in which the bias inherent to NRE-B is discouraged by the loss function and it does not appear at optimum. Figure 1 diagrams the interpolated performance as a function of hyperparameters. It shows which settings recover NRE-A and NRE-B, also indicating that highest performance occurs with settings distant from these. Figure 2 shows the relationship of the loss functions. We call our framework NRE-C[1] and expound the details in Section 2.

An existing importance sampling diagnostic [30] tests whether a classifier can distinguish $p(\boldsymbol{x}\,|\,\boldsymbol{\theta})$ samples from from samples from $p(\boldsymbol{x})$ weighted by the estimated ratio. We demonstrate that, when estimating accurate posteriors, our proposed NRE-C passes this diagnostic while NRE-B does not.

---

[1]The code for our project can be found at https://github.com/bkmi/cnre under the Apache License 2.0.

Taking inspiration from mutual information estimation [58], we propose applying a variational bound on the mutual information between $\boldsymbol{\theta}$ and $\boldsymbol{x}$ in a novel way–as an informative metric measuring a lower bound on the Kullback-Leibler divergence between surrogate posterior estimate $p_{\boldsymbol{w}}(\boldsymbol{\theta} \,|\, \boldsymbol{x})$ and $p(\boldsymbol{\theta} \,|\, \boldsymbol{x})$, averaged over $p(\boldsymbol{x})$. Unlike with two-sample testing methods commonly used in machine learning literature [44], our metric samples only from $p(\boldsymbol{\theta}, \boldsymbol{x})$, which is always available in SBI, and does not require samples from the intractable $p(\boldsymbol{\theta} \,|\, \boldsymbol{x})$. Our metric is meaningful to scientists working on problems with intractable posteriors. The technique requires estimating the partition function, which can be expensive. We find the metric to be well correlated with results from two-sample tests.

We evaluate NRE-B and NRE-C in a fair comparison in several training regimes in Section 3. We perform a hyperparameter search on three simulators with tractable likelihood by benchmarking the behavior when (a) jointly drawn pairs $(\boldsymbol{\theta}, \boldsymbol{x})$ are unlimited or when jointly drawn pairs $(\boldsymbol{\theta}, \boldsymbol{x})$ are fixed but we (b) can draw from the prior $p(\boldsymbol{\theta})$ without limit or (c) are restricted to the initial pairs. We also perform the SBI benchmark of Lueckmann et al. [44] with our recommended hyperparameters.

## 2 Methods

The ratio between probability distributions can be estimated using the "likelihood ratio trick" by training a classifier to distinguish samples [12, 19, 27, 30, 50, 66, 69]. We first summarize the loss functions of NRE-A and NRE-B which approximate the *intractable* likelihood-to-evidence ratio $r(\boldsymbol{x} \,|\, \boldsymbol{\theta}) \coloneqq \frac{p(\boldsymbol{x} \,|\, \boldsymbol{\theta})}{p(\boldsymbol{x})}$. We then elaborate on our proposed generalization, NRE-C. Finally, we explain how to recover NRE-A and NRE-B within our framework and comment on the normalization properties.

**NRE-A** Hermans et al. [30] train a binary classifier to distinguish $(\boldsymbol{\theta}, \boldsymbol{x})$ pairs drawn dependently $p(\boldsymbol{\theta}, \boldsymbol{x})$ from those drawn independently $p(\boldsymbol{\theta})p(\boldsymbol{x})$. This classifier is parameterized by a neural network $f_{\boldsymbol{w}}$ which approximates $\log r(\boldsymbol{x} \,|\, \boldsymbol{\theta})$. We seek optimal network weights

$$\boldsymbol{w}^* \in \underset{\boldsymbol{w}}{\arg\min} -\frac{1}{2B} \left[ \sum_{b=1}^{B} \log\left(1 - \sigma \circ f_{\boldsymbol{w}}(\boldsymbol{\theta}^{(b)}, \boldsymbol{x}^{(b)})\right) + \sum_{b'=1}^{B} \log\left(\sigma \circ f_{\boldsymbol{w}}(\boldsymbol{\theta}^{(b')}, \boldsymbol{x}^{(b')})\right) \right] \quad (3)$$

$\boldsymbol{\theta}^{(b)}, \boldsymbol{x}^{(b)} \sim p(\boldsymbol{\theta})p(\boldsymbol{x})$ and $\boldsymbol{\theta}^{(b')}, \boldsymbol{x}^{(b')} \sim p(\boldsymbol{\theta}, \boldsymbol{x})$ over $B$ samples. NRE-A's ratio estimate converges to $f_{\boldsymbol{w}^*} = \log \frac{p(\boldsymbol{x} \,|\, \boldsymbol{\theta})}{p(\boldsymbol{x})}$ given unlimited model flexibility and data. Details can be found in Appendix A.

**NRE-B** Durkan et al. [16] train a classifier that selects from among $K$ parameters $(\boldsymbol{\theta}_1, \ldots, \boldsymbol{\theta}_K)$ which could have generated $\boldsymbol{x}$, in contrast with NRE-A's binary possibilities. One of these parameters $\boldsymbol{\theta}_k$ is *always* drawn jointly with $\boldsymbol{x}$. The classifier is parameterized by a neural network $g_{\boldsymbol{w}}$ which approximates $\log r(\boldsymbol{x} \,|\, \boldsymbol{\theta})$. Training is done over $B$ samples by finding

$$\boldsymbol{w}^* \in \underset{\boldsymbol{w}}{\arg\min} \left[ -\frac{1}{B} \sum_{b'=1}^{B} \log \frac{\exp \circ g_{\boldsymbol{w}}(\boldsymbol{\theta}_k^{(b')}, \boldsymbol{x}^{(b')})}{\sum_{i=1}^{K} \exp \circ g_{\boldsymbol{w}}(\boldsymbol{\theta}_i^{(b')}, \boldsymbol{x}^{(b')})} \right] \quad (4)$$

where $\boldsymbol{\theta}_1^{(b')}, \ldots, \boldsymbol{\theta}_K^{(b')} \sim p(\boldsymbol{\theta})$ and $\boldsymbol{x}^{(b')} \sim p(\boldsymbol{x} \,|\, \boldsymbol{\theta}_k^{(b')})$. Given unlimited model flexibility and data NRE-B's ratio estimate converges to $g_{\boldsymbol{w}^*}(\boldsymbol{\theta}, \boldsymbol{x}) = \log \frac{p(\boldsymbol{\theta} \,|\, \boldsymbol{x})}{p(\boldsymbol{\theta})} + c_{\boldsymbol{w}^*}(\boldsymbol{x})$. Details are in Appendix A.

### 2.1 Contrastive Neural Ratio Estimation

Our proposed algorithm NRE-C trains a classifier to identify which $\boldsymbol{\theta}$ among $K$ candidates is responsible for generating a given $\boldsymbol{x}$, inspired by NRE-B. We added another option that indicates $\boldsymbol{x}$ was drawn independently, inspired by NRE-A. The introduction of the additional class yields a ratio without the specific $c_{\boldsymbol{w}}(\boldsymbol{x})$ bias at optimum. Define $\boldsymbol{\Theta} \coloneqq (\boldsymbol{\theta}_1, ..., \boldsymbol{\theta}_K)$ and conditional probability

$$p_{\text{NRE-C}}(\boldsymbol{\Theta}, \boldsymbol{x} \,|\, y = k) \coloneqq \begin{cases} p(\boldsymbol{\theta}_1) \cdots p(\boldsymbol{\theta}_K) p(\boldsymbol{x}) & k = 0 \\ p(\boldsymbol{\theta}_1) \cdots p(\boldsymbol{\theta}_K) p(\boldsymbol{x} \,|\, \boldsymbol{\theta}_k) & k = 1, \ldots, K \end{cases}. \quad (5)$$

We set marginal probabilities $p(y = k) := p_K$ for all $k \geq 1$ and $p(y = 0) := p_0$, yielding the relationship $p_0 = 1 - Kp_K$. Let the odds of any pair being drawn dependently to completely independently be $\gamma := \frac{Kp_K}{p_0}$. We now use Bayes' formula to compute the conditional probability

$$
\begin{aligned}
p(y = k \mid \boldsymbol{\Theta}, \boldsymbol{x}) &= \frac{p(y = k)\, p(\boldsymbol{\Theta}, \boldsymbol{x} \mid y = k)/p(\boldsymbol{\Theta}, \boldsymbol{x} \mid y = 0)}{\sum_{i=0}^{K} p(y = i)\, p(\boldsymbol{\Theta}, \boldsymbol{x} \mid y = i)/p(\boldsymbol{\Theta}, \boldsymbol{x} \mid y = 0)} \\
&= \frac{p(y = k)\, p(\boldsymbol{\Theta}, \boldsymbol{x} \mid y = k)/p(\boldsymbol{\Theta}, \boldsymbol{x} \mid y = 0)}{p(y = 0) + \sum_{i=1}^{K} p(y = i)\, p(\boldsymbol{\Theta}, \boldsymbol{x} \mid y = i)/p(\boldsymbol{\Theta}, \boldsymbol{x} \mid y = 0)} \\
&= \begin{cases} \frac{K}{K + \gamma \sum_{i=1}^{K} r(\boldsymbol{x} \mid \boldsymbol{\theta}_i)} & k = 0 \\ \frac{\gamma\, r(\boldsymbol{x} \mid \boldsymbol{\theta}_k)}{K + \gamma \sum_{i=1}^{K} r(\boldsymbol{x} \mid \boldsymbol{\theta}_i)} & k = 1, \dots, K \end{cases}.
\end{aligned}
\tag{6}
$$

We dropped the NRE-C subscript and substituted in $\gamma$ to replace the $p(y)$ class probabilities. We train a classifier, parameterized by neural network $h_{\boldsymbol{w}}(\boldsymbol{\theta}, \boldsymbol{x})$ with weights $\boldsymbol{w}$, to approximate (6) by

$$
q_{\boldsymbol{w}}(y = k \mid \boldsymbol{\Theta}, \boldsymbol{x}) = \begin{cases} \frac{K}{K + \gamma \sum_{i=1}^{K} \exp \circ h_{\boldsymbol{w}}(\boldsymbol{\theta}_i, \boldsymbol{x})} & k = 0 \\ \frac{\gamma\, \exp \circ h_{\boldsymbol{w}}(\boldsymbol{\theta}_k, \boldsymbol{x})}{K + \gamma \sum_{i=1}^{K} \exp \circ h_{\boldsymbol{w}}(\boldsymbol{\theta}_i, \boldsymbol{x})} & k = 1, \dots, K. \end{cases}
\tag{7}
$$

We note that (7) still satisfies $\sum_{k=0}^{K} q_{\boldsymbol{w}}(y = k \mid \boldsymbol{\Theta}, \boldsymbol{x}) = 1$, no matter the parameterization.

**Optimization** We design a loss function that encourages $h_{\boldsymbol{w}}(\boldsymbol{\theta}, \boldsymbol{x}) = \log \frac{p(\boldsymbol{x} \mid \boldsymbol{\theta})}{p(\boldsymbol{x})}$ at convergence, and holds at optimum with unlimited flexibility and data. We introduce the cross entropy loss

$$
\begin{aligned}
\ell(\boldsymbol{w}) &:= \mathbb{E}_{p(y, \boldsymbol{\Theta}, \boldsymbol{x})} \left[ -\log q_{\boldsymbol{w}}(y \mid \boldsymbol{\Theta}, \boldsymbol{x}) \right] \\
&= -p_0 \mathbb{E}_{p(\boldsymbol{\Theta}, \boldsymbol{x} \mid y=0)} \left[ \log q_{\boldsymbol{w}}(y = 0 \mid \boldsymbol{\Theta}, \boldsymbol{x}) \right] - p_K \sum_{k=1}^{K} \mathbb{E}_{p(\boldsymbol{\Theta}, \boldsymbol{x} \mid y=k)} \left[ \log q_{\boldsymbol{w}}(y = k \mid \boldsymbol{\Theta}, \boldsymbol{x}) \right] \\
&= -p_0 \mathbb{E}_{p(\boldsymbol{\Theta}, \boldsymbol{x} \mid y=0)} \left[ \log q_{\boldsymbol{w}}(y = 0 \mid \boldsymbol{\Theta}, \boldsymbol{x}) \right] - Kp_K \mathbb{E}_{p(\boldsymbol{\Theta}, \boldsymbol{x} \mid y=K)} \left[ \log q_{\boldsymbol{w}}(y = K \mid \boldsymbol{\Theta}, \boldsymbol{x}) \right]
\end{aligned}
\tag{8}
$$

and minimize it towards $\boldsymbol{w}^* \in \arg\min_{\boldsymbol{w}} \ell(\boldsymbol{w})$. We point out that the final term is symmetric up to permutation of $\boldsymbol{\Theta}$, enabling the replacement of the sum by multiplication with $K$. When $\gamma$ and $K$ are known, $p_0 = \frac{1}{1+\gamma}$ and $p_K = \frac{1}{K} \frac{\gamma}{1+\gamma}$ under our constraints. Without loss of generality, we let $\boldsymbol{\theta}_1, \dots, \boldsymbol{\theta}_K \sim p(\boldsymbol{\theta})$ and $\boldsymbol{x} \sim p(\boldsymbol{x} \mid \boldsymbol{\theta}_K)$. An empirical estimate of the loss on $B$ samples is therefore

$$
\begin{aligned}
\hat{\ell}_{\gamma, K}(\boldsymbol{w}) := -\frac{1}{B} \Bigg[ &\frac{1}{1 + \gamma} \sum_{b=1}^{B} \log q_{\boldsymbol{w}} \left( y = 0 \mid \boldsymbol{\Theta}^{(b)}, \boldsymbol{x}^{(b)} \right) \\
&+ \frac{\gamma}{1 + \gamma} \sum_{b'=1}^{B} \log q_{\boldsymbol{w}} \left( y = K \mid \boldsymbol{\Theta}^{(b')}, \boldsymbol{x}^{(b')} \right) \Bigg].
\end{aligned}
\tag{9}
$$

In the first term, the classifier sees a completely independently drawn sample of $\boldsymbol{x}$ and $\boldsymbol{\Theta}$ while $\boldsymbol{\theta}_K$ is drawn jointly with $\boldsymbol{x}$ in the second term. In both terms, the classifier considers $K$ choices. In practice, we bootstrap both $\boldsymbol{\theta}_1^{(b)}, \dots, \boldsymbol{\theta}_K^{(b)}$ and $\boldsymbol{\theta}_1^{(b')}, \dots, \boldsymbol{\theta}_{K-1}^{(b')}$ from the same mini-batch and compare them to the same $\boldsymbol{x}$, similarly to NRE-A and NRE-B. Proof of the above is in Appendix B.

**Recovering NRE-A and NRE-B** NRE-C is general because specific hyperparameter settings recover NRE-A and NRE-B. To recover NRE-A one should set $\gamma = 1$ and $K = 1$ in (9) yielding

$$
\begin{aligned}
\hat{\ell}_{1,1}(\boldsymbol{w}) &= -\frac{1}{2B} \left[ \sum_{b=1}^{B} \log \frac{1}{1 + \exp \circ h_{\boldsymbol{w}}(\boldsymbol{\theta}^{(b)}, \boldsymbol{x}^{(b)})} + \sum_{b'=1}^{B} \log \frac{\exp \circ h_{\boldsymbol{w}}(\boldsymbol{\theta}^{(b')}, \boldsymbol{x}^{(b')})}{1 + \exp \circ h_{\boldsymbol{w}}(\boldsymbol{\theta}^{(b')}, \boldsymbol{x}^{(b')})} \right] \\
&= -\frac{1}{2B} \left[ -\sum_{b=1}^{B} \log \left( 1 - \sigma \circ h_{\boldsymbol{w}}(\boldsymbol{\theta}^{(b)}, \boldsymbol{x}^{(b)}) \right) + \sum_{b'=1}^{B} \log \left( \sigma \circ h_{\boldsymbol{w}}(\boldsymbol{\theta}^{(b')}, \boldsymbol{x}^{(b')}) \right) \right]
\end{aligned}
\tag{10}
$$

where we dropped the lower index. Recovering NRE-B requires taking the limit $\gamma \to \infty$ in the loss function. In that case, the first term goes to zero, and second term converges to the softmax function.

$$\hat{\ell}_{\infty,K}(\boldsymbol{w}) = \lim_{\gamma \to \infty} \hat{\ell}_{\gamma,K}(\boldsymbol{w}) \quad = -\frac{1}{B}\left[ \sum_{b'=1}^{B} \log \frac{\exp \circ h_{\boldsymbol{w}}(\boldsymbol{\theta}_k, \boldsymbol{x}))}{\sum_{i=1}^{K} \exp \circ h_{\boldsymbol{w}}(\boldsymbol{\theta}_i, \boldsymbol{x})} \right] \tag{11}$$

is determined by substitution into (9). Both equations are obviously the same as their counterparts.

**Estimating a normalized posterior**   In the limit of infinite data and infinite neural network capacity (width, depth) the optimal classifier trained using NRE-C (with $\gamma \in \mathbb{R}^+$) satisfies the equality:

$$h_{\boldsymbol{w}^*}(\boldsymbol{\theta}, \boldsymbol{x}) = \log \frac{p(\boldsymbol{\theta} \mid \boldsymbol{x})}{p(\boldsymbol{\theta})}. \tag{12}$$

In particular, we have that the following normalizing constant is trivial:

$$Z(\boldsymbol{x}) := \int \exp\left(h_{\boldsymbol{w}^*}(\boldsymbol{\theta}, \boldsymbol{x})\right) p(\boldsymbol{\theta})\, d\boldsymbol{\theta} = \int p(\boldsymbol{\theta} \mid \boldsymbol{x})\, d\boldsymbol{\theta} = 1. \tag{13}$$

This is a result of Lemma 1 in Appendix B. However, practitioners never operate in this setting, rather they use finite sample sizes and neural networks with limited capacity that are optimized locally. The non-optimal function $\exp(h_{\boldsymbol{w}}(\boldsymbol{\theta}, \boldsymbol{x}))$ does not have a direct interpretation as a ratio of probability distributions, rather as the function to weigh the prior $p(\boldsymbol{\theta})$ to approximate the unnormalized posterior. In other words, we find the following approximation for the posterior $p(\boldsymbol{\theta} \mid \boldsymbol{x})$:

$$p_{\boldsymbol{w}}(\boldsymbol{\theta} \mid \boldsymbol{x}) := \frac{\exp(h_{\boldsymbol{w}}(\boldsymbol{\theta}, \boldsymbol{x}))}{Z_{\boldsymbol{w}}(\boldsymbol{x})} p(\boldsymbol{\theta}), \qquad Z_{\boldsymbol{w}}(\boldsymbol{x}) := \int \exp\left(h_{\boldsymbol{w}}(\boldsymbol{\theta}, \boldsymbol{x})\right) p(\boldsymbol{\theta})\, d\boldsymbol{\theta}, \tag{14}$$

where in general the normalizing constant is not trivial, i.e. $Z_{\boldsymbol{w}}(\boldsymbol{x}) \neq 1$. As stated above, the NRE-C (and NRE-A) objective encourages $Z_{\boldsymbol{w}}(\boldsymbol{x})$ to converge to 1. This is in sharp contrast to NRE-B, where even at optimum with an unrestricted function class a non-trivial $\boldsymbol{x}$-dependent bias term can appear.

There is no restriction on how pathological the NRE-B bias $c_{\boldsymbol{w}}(\boldsymbol{x})$ can be. Consider a minimizer of (4), the NRE-B loss function, $h_{\boldsymbol{w}^*} + c_{\boldsymbol{w}^*}(\boldsymbol{x})$. Adding any function $d(\boldsymbol{x})$ cancels out in the fraction and is also a minimizer of (4). This freedom complicates any numerical computation of the normalizing constant and renders the importance sampling diagnostic from Section 2.2 generally inapplicable. We report Monte Carlo estimates of $Z_{\boldsymbol{w}}(\boldsymbol{x})$ on a test problem across hyperparameters in Figure 14.

## 2.2   Measuring performance & ratio estimator diagnostics

SBI is difficult to verify because, for many use cases, the practitioner cannot compare surrogate $p_{\boldsymbol{w}}(\boldsymbol{\theta} \mid \boldsymbol{x})$ to the intractable ground truth $p(\boldsymbol{\theta} \mid \boldsymbol{x})$. Incongruous with the practical use case for SBI, much of the literature has focused on measuring the similarity between surrogate and posterior using two-samples tests on tractable problems. For comparison with literature, we first reference a two-sample exactness metric which requires a tractable posterior. We then discuss diagnostics which do not require samples from $p(\boldsymbol{\theta} \mid \boldsymbol{x})$, commenting on the relevance for each NRE algorithm with empirical results. Further, we find that a known variational bound to the mutual information is tractable to estimate within SBI, that it bounds the average Kullback-Leibler divergence between surrogate and posterior, and propose to use it for model comparison on intractable inference tasks.

**Comparing to a tractable posterior with estimates of exactness**   Assessments of approximate posterior quality are available when samples can be drawn from both the posterior $\boldsymbol{\theta} \sim p(\boldsymbol{\theta} \mid \boldsymbol{x})$ and the approximation $\boldsymbol{\theta} \sim q(\boldsymbol{\theta} \mid \boldsymbol{x})$. In the deep learning-based SBI literature, exactness is measured as a function of computational cost, usually simulator calls. We investigate this with NRE-C in Section 3.3.

Based on the recommendations of Lueckmann et al. [44] our experimental results are measured using the Classifier Two-Sample Test (C2ST) [17, 40, 41]. A classifier is trained to distinguish samples from either the surrogate or the ground truth posterior. An average classification probability on holdout data of 1.0 implies that samples from each distribution are easily identified; 0.5 implies either the distributions are the same or the classifier does not have the capacity to distinguish them.

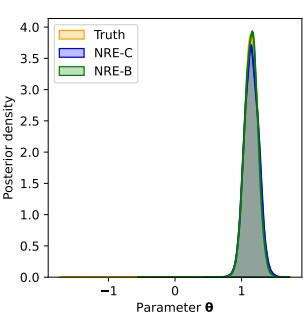

(a) Posteriors

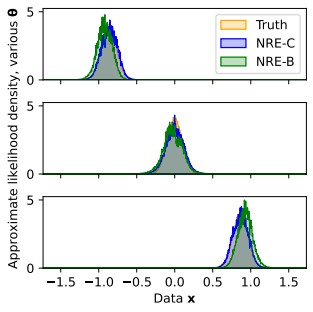

(b) Likelihood and reweighted marginal generative models.

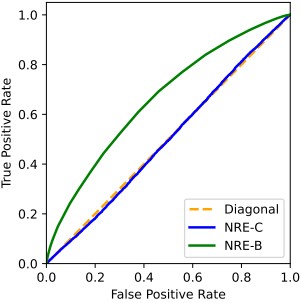

(c) Receiver operating characteristic.

Figure 3: The figures visualize the importance sampling diagnostic on ratio estimators trained using NRE-B and NRE-C. (a) Both methods produce satisfactory posterior estimates that agree with $p(\boldsymbol{\theta} \,|\, \boldsymbol{x})$. (b) $p(\boldsymbol{x} \,|\, \boldsymbol{\theta})$ is shown along with $p(\boldsymbol{x})$ samples weighted by NRE-A $\exp \circ f_{\boldsymbol{w}}(\boldsymbol{\theta}, \boldsymbol{x})$ and NRE-B $\exp \circ g_{\boldsymbol{w}}(\boldsymbol{\theta}, \boldsymbol{x})$. Each plot corresponds to a different $\boldsymbol{\theta}$. Despite high posterior accuracy, the NRE-B estimates are distinct from $p(\boldsymbol{x} \,|\, \boldsymbol{\theta})$. (c) Two classifier's ROC curves, each trained to distinguish $p(\boldsymbol{x} \,|\, \boldsymbol{\theta})$ samples from $p(\boldsymbol{x})$ samples weighted by the corresponding NRE's $\hat{r}$ estimate. The classifier failed to distinguish likelihood samples from the NRE-C weighted data samples, but successfully identified NRE-B weighted samples. NRE-B accurately approximates the posterior, but fails the diagnostic. NRE-C produces an accurate posterior surrogate and passes the diagnostic.

**Importance sampling diagnostic**  An accurate likelihood-to-evidence weight transforms the data distribution into the likelihood by $p(\boldsymbol{x} \,|\, \boldsymbol{\theta}) = p(\boldsymbol{x}) r(\boldsymbol{x} \,|\, \boldsymbol{\theta})$. Since NRE necessitates simulator access, we can test the ratio estimator by training a classifier to distinguish unweighted $p(\boldsymbol{x} \,|\, \boldsymbol{\theta})$ samples from weighted $p(\boldsymbol{x}) \hat{r}(\boldsymbol{x} \,|\, \boldsymbol{\theta})$ samples, where $\hat{r}$ implies an estimate. Indistinguishability between samples implies either that the approximate ratio is accurate for parameter $\boldsymbol{\theta}$ or that the classifier does not have sufficient power to find predictive features. Issues with classification power can be detected by assessing the classifier's ability to distinguish $p(\boldsymbol{x})$ from $p(\boldsymbol{x} \,|\, \boldsymbol{\theta})$. The performance can be visualized in a receiver operating curve (ROC) or measured by the area under the curve (ROC AUC). This diagnostic has been used for ratio estimators before [12, 30] but it comes from training models under covariate shift [62]. It is particularly appealing because it does not require samples from $p(\boldsymbol{\theta} \,|\, \boldsymbol{x})$.

Durkan et al. [16] do not mention this diagnostic in their paper, but due to its intrinsic bias NRE-B does not fulfill the identity necessary for this diagnostic to hold at optimum. The unknown factor that depends on $\boldsymbol{x}$ implies $p(\boldsymbol{x} \,|\, \boldsymbol{\theta}) \neq p(\boldsymbol{x}) \exp \circ g_{\boldsymbol{w}}(\boldsymbol{x} \,|\, \boldsymbol{\theta})$. We provide empirical evidence of this issue in Figure 3. Although NRE-B accurately approximates the true posterior, it demonstrably fails the diagnostic. Given the limited options for verification of SBI results, this presents a major problem by significantly limiting the trustworthiness of NRE-B on any problem with an intractable posterior. In Appendix B, we show that the unrestricted NRE-B-specific $c_{\boldsymbol{w}}(\boldsymbol{x})$ bias means approximating $p(\boldsymbol{x} \,|\, \boldsymbol{\theta})$ with normalized importance weights will not solve the issue.

**Mutual information bound**  Selecting the surrogate model most-similar to the target posterior remains intractable without access to $p(\boldsymbol{\theta} \,|\, \boldsymbol{x}_o)$. Nevertheless, practitioners must decide which surrogate should approximate the posterior across training and hyperparameter search. Unfortunately, the validation losses between different versions of NRE and different $K$ and $\gamma$ settings are not comparable. A good heuristic is to choose the model which minimizes the Kullback-Leibler divergence *on average* over possible data $p(\boldsymbol{x})$. In Appendix D, we prove the relationship between $I(\boldsymbol{\theta}; \boldsymbol{x})$, the *mutual information* with respect to $p(\boldsymbol{\theta}, \boldsymbol{x})$, our models' variational bound $I_{\boldsymbol{w}}^{(0)}(\boldsymbol{\theta}; \boldsymbol{x})$, and the average KLD

$$\mathbb{E}_{p(\boldsymbol{x})}\left[\mathrm{KLD}(p(\boldsymbol{\theta} \,|\, \boldsymbol{x}) \,\|\, p_{\boldsymbol{w}}(\boldsymbol{\theta} \,|\, \boldsymbol{x}))\right] = I(\boldsymbol{\theta}; \boldsymbol{x}) - I_{\boldsymbol{w}}^{(0)}(\boldsymbol{\theta}; \boldsymbol{x}), \tag{15}$$

$$I_{\boldsymbol{w}}^{(0)}(\boldsymbol{\theta}; \boldsymbol{x}) \coloneqq \mathbb{E}_{p(\boldsymbol{\theta}, \boldsymbol{x})}\left[\log \hat{r}(\boldsymbol{x} \,|\, \boldsymbol{\theta})\right] - \mathbb{E}_{p(\boldsymbol{x})}\left[\log \mathbb{E}_{p(\boldsymbol{\theta})}[\hat{r}(\boldsymbol{x} \,|\, \boldsymbol{\theta})]\right]. \tag{16}$$

The non-negativity of all terms in (15) implies $I(\boldsymbol{\theta}; \boldsymbol{x}) \geq I_{\boldsymbol{w}}^{(0)}(\boldsymbol{\theta}; \boldsymbol{x})$; that means the model which minimizes $-I_{\boldsymbol{w}}^{(0)}(\boldsymbol{\theta}; \boldsymbol{x})$ best satisfies our heuristic. We propose to approximate $-I_{\boldsymbol{w}}^{(0)}(\boldsymbol{\theta}; \boldsymbol{x})$ with Monte Carlo using held-out data as a metric for model selection during training and across hyperparameters. The expectation values average over $p(\boldsymbol{\theta}, \boldsymbol{x})$, $p(\boldsymbol{\theta})$, and $p(\boldsymbol{x})$. We can sample from all

of these distributions in the SBI context. Since the second term in (16) computes the average log partition function, our metric can compare NRE-B-based surrogates to NRE-C-based ones. However, the metric comes with the normal challenges of estimating the log partition function, which can be very expensive. Additionally, the presence of the ratio in the integrand can make this integral high variance. We treat a generally tractable bound in Appendix D. We go into *much* more depth there and discuss of the relevance to Neural Posterior Estimation [16, 21, 42, 53]. While the application to SBI is novel, bounds on the mutual information have been broadly investigated for contrastive representation learning and mutual information estimation [4, 22, 23, 25, 58, 71].

**Empirical expected coverage probability**  For a candidate distribution to qualify as the posterior, integrating over data must return the prior. A measurement that follows from calibration to the prior is called expected coverage probability. Expected coverage probability can be estimated with samples from $p(\boldsymbol{\theta}, \boldsymbol{x})$ and any amortized SBI method. Although important, ability to compute this metric does not distinguish NRE-C. We refer the interested reader to Hermans et al. [31]. We note that popular sequential techniques generally render this diagnostic inapplicable, with exceptions [10, 47, 48].

## 3  Experiments

We perform experiments in three settings to measure the exactness of surrogate models under various hyperparameters and training regimes. Section 3.1 aims to identify whether data or architecture is the bottleneck in accuracy by drawing from the joint with every mini-batch. The next experiments aim to determine how to optimally extract inference information given a limited amount of simulation data. Section 3.2 leverages a cheap prior by drawing new contrastive parameters with every mini-batch. Finally, Section 3.3 applies the commonplace training regime for deep learning-based SBI literature of fixed data and bootstrapped contrastive parameters. In this setting, we also benchmark NRE-C on ten inference tasks from Lueckmann et al. [44] using our recommended hyperparameters.

On all hyperparameter searches we consider three simulators from the simulation-based inference benchmark, namely SLCP, Two Moons, and Gaussian Mixture [44]. SLCP is a difficult inference task which has a simple likelihood and complex posterior [21, 56]. Parameters are five dimensional and the data are four samples from a two dimensional random variable. Two Moons is two dimensional with a crescent-shaped bimodal posterior. Finally, the Gaussian Mixture data draws from a mixture of two, two-dimensional normal distributions with extremely different covariances [3, 63, 64, 70].

Our surrogate models are parameterized by one of these architectures: *Small NN* is like the benchmark with 50 hidden units and two residual blocks. *Large NN* has 128 hidden units and three residual blocks. We use batch normalization, unlike the benchmark. We compare their performance on a grid of $\gamma$ and $K$ values. We report post-training results using the C2ST, and mid-training validation loss for insight into convergence rate. We generally use residual networks [28] with batch normalization [33] and train them using adam [37]. We also run NRE-B with the same architecture for comparison. To compare with NRE-B we set the number of total contrastive parameters equal.

What does fair comparison mean in our experimentation? We compare models across fixed number of gradient steps. This implies that models with more classes, i.e., greater $K$, evaluate the classifier-in-training on more pairs at a given training step than a model with fewer classes. An alternative which we do *not* apply in this paper: Vary the number of gradient steps, holding the number of pair evaluations fixed, i.e., a model with higher $K$ sees the same number of pairs as a model with lower $K$ but the model with lower $K$ has been updated more times. We leave this analysis for future work.

### 3.1  Behavior with unlimited data

What is responsible for inaccuracies of the surrogate model when training has saturated? (a) the amount of training data (b) the flexibility of the model? In this section we provide new simulation and parameter data with every mini-batch and train until saturation, thereby eliminating the possibility of (a). The newly drawn mini-batch parameters $\boldsymbol{\Theta}$ are bootstrapped for the contrastive pairs.

The setting is similar to REJ-ABC where simulations are drawn until the posterior has converged. The results of this study will provide a baseline to compare with limited-data results and help us understand how the deep learning architecture's limitations are affected by our introduced hyperparameters. The results are reported in the top row of Figure 5.

The trend is that increasing the number of contrastive examples helps NRE-B and NRE-C. There is not a clear trend with respect to $\gamma$ in the C2ST. Study of the detailed validation losses in Appendix C reveals that high $\gamma$ is associated with higher variance validation loss during training, but here it does not seem to strongly affect convergence rate. Saturation is reached before the maximum number of epochs have elapsed and Large NN converges to a better result. Although obfuscated in the averaged results in Figure 5, the result is obvious on SLCP, the most difficult inference task that provides an opportunity for a more flexible architecture to improve.

We argue that generally the performance bottleneck has to do with the number of contrastive parameters, network flexibility, or training details rather than the amount of data. Despite saturating validation losses, the C2ST improves based on these factors. Intuitively a more flexible model could continue to extract information from this unlimited set of jointly drawn $(\boldsymbol{\theta}, \boldsymbol{x})$ pairs. This has some consequence because network flexibility may limit performance in the benchmark case. Appendix C contains more detailed results.

## 3.2 Leveraging fast priors (drawing theta)

In practice, the simulator is often slow but one can often draw from the prior at the same pace as training a neural network for one mini-batch. Our goal is to understand how our hyperparameter are affected by this setting and whether it is valuable to use this technique in practice. To our knowledge, this training regime has not been explored in the deep learning-based SBI literature.

Initially we draw a fixed set of around 20,000 samples $(\boldsymbol{\theta}, \boldsymbol{x}) \sim p(\boldsymbol{\theta}, \boldsymbol{x})$. For every mini-batch during training, we sample all necessary contrastive parameters from the prior. For each term in $\hat{\ell}_{\gamma,K}$ we take the same batch of contrastive parameters and reshuffle them, thereby equalizing the number of samples seen by NRE-B and NRE-C, up to bootstrap. The averaged C2ST results from the three inference tasks are reported in the middle row of Figure 5.

The resulting estimators are markedly less sensitive to the number of contrastive parameters than in Section 3.1. The number of network parameters has a positive effect, although again this is best seen on the hardest task, SLCP, in the Appendix C. Empirically, lower values of $\gamma$ improve performance with Large NN; however, there is still high variance and this result may be from noise. In this setting, $\gamma \neq 1$ has a very small negative effect on convergence rate compared to unity, as seen in the validation loss plots in Appendix C. Since training has saturated, we claim this result implies drawing contrastive parameters from the prior helps extract the maximum amount of information from fixed simulation data $\boldsymbol{x}$–without being strongly affected by the number of contrastive parameters $K$.

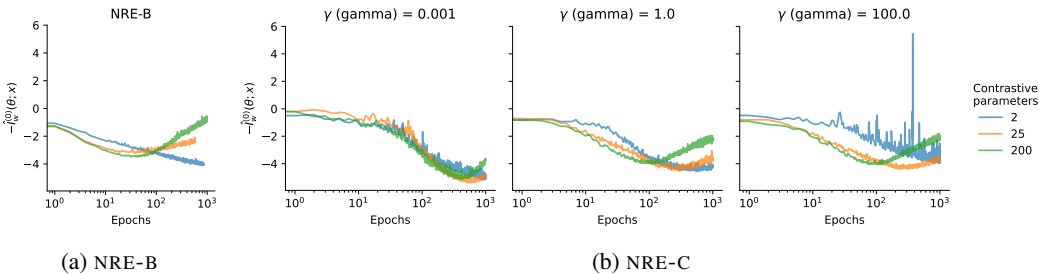

(a) NRE-B                    (b) NRE-C

Figure 4: Our proposed metric, a negative bound on the mutual information $-\hat{I}_{\boldsymbol{w}}^{(0)}(\boldsymbol{\theta}; \boldsymbol{x})$, for the SLCP task estimated over the validation set versus training epochs using (a) NRE-B and (b) NRE-C with various values of $\gamma$ and $K$, a Large NN architecture, and fixed training data. The bound permits visualization of the convergence rates and pairwise comparison across models. When $K$ is fixed, small $\gamma$ slows convergences but leads to better optima than large $\gamma$. When $\gamma$ is fixed, small $K$ slows convergence but leads to better optima than high $K$. Unlike when computing the validation loss directly, this metric does not exhibit a $\gamma$ and $K$ dependent bias as seen in Figures 7, 8, and 9.

## 3.3 Simulation-based inference benchmark

In this section, we assume the traditional literature setting of limited simulation and prior budget. Once we've selected hyperparameters based on a grid search of a subset of the SBI benchmark, we perform the entire benchmark with those hyperparameters.

**Hyperparameter search** To compare to the previous two sections, the amount of training data was fixed such that each epoch was comparable to Section 3.1, this amounts to about 20,000 samples. We first inspect the C2ST results in the last row of Figure 5. The sensitivity to $K$ appears less than in Section 3.1 but more than in Section 3.2. Smaller $\gamma$ slightly improves Large NN's performance but the noise makes this result uncertain. The larger network performs better, see Appendix C.

We do a Monte Carlo estimate of our proposed metric, denoted $-\hat{I}_w^{(0)}(\boldsymbol{\theta}; \boldsymbol{x})$, to show performance on the SLCP task as a function of epochs in Figure 4. For both NRE-B and NRE-C, increasing $K$ tends to positively affect the convergence rate. Generally, with fixed $\gamma$ and high $K$, peak performance is negatively impacted, in contrast to results from Figure 5 using the C2ST. Generally, with fixed $K$, small $\gamma$ slows convergences but leads to better optima than large $\gamma$. When $\gamma$ is fixed, small $K$ slows convergence but leads to better optima than high $K$. See Figure 12.

Our take-away is that $\gamma \in [0.1, 10]$ has a small effect, otherwise learning can become unstable or very slow. $\gamma = 1$ has a good compromise on high convergence rate without sacrificing too much performance. Generally, we saw improved performance on the C2ST by increasing contrastive parameters $K$. However, larger $K$ also increases the magnitude of the normalizing constant, and extreme values reduced the performance on $-\hat{I}_w^{(0)}(\boldsymbol{\theta}; \boldsymbol{x})$, see Appendix D. Since the benchmark compares C2ST, we optimized our architecture based on that metric. Due to bootstrapping parameters from the batched $\boldsymbol{\theta}$, the maximum $K$ is $B/2$ without reusing any $\boldsymbol{\theta}$ to compare with $\boldsymbol{x}$. This is considering both terms in our loss function.

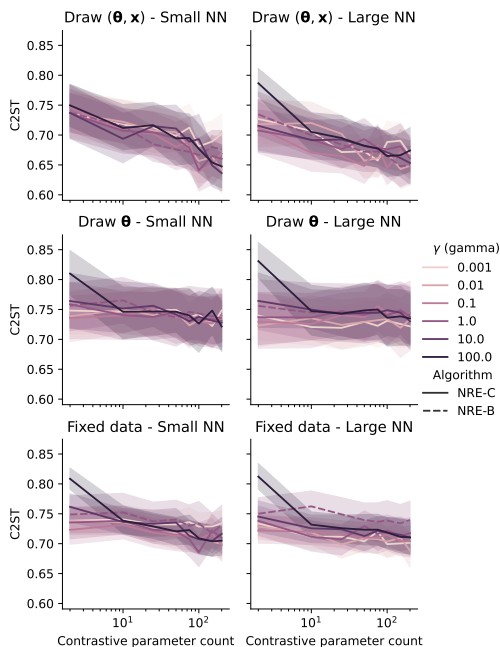

Figure 5: Exactness of NRE posterior surrogates is computed for various contrastive parameter counts, $\gamma$ values, and architectures on an *average* of three tasks from the SBI benchmark [44]. C2ST assigns 1.0 to inaccurate and 0.5 to accurate posterior approximation. (top) $p(\boldsymbol{\theta}, \boldsymbol{x})$ was sampled at every mini-batch during training. The accuracy strongly depends on $K$. (mid) A fixed number of dependent $(\boldsymbol{\theta}, \boldsymbol{x})$ pairs were drawn, but $p(\boldsymbol{\theta})$ was sampled at every mini-batch during training. In this regime, $K$ has a smaller effect. (bot) The training data is completely fixed. Contrastive parameters are drawn in a bootstrap from the mini-batch. On the problems with fixed simulation data $\boldsymbol{x}$, higher $K$ improves accuracy and small $\gamma$ with larger architectures slightly improves performance. The effects of the architecture are more clearly seen on difficult problems like SLCP in Appendix C.

**Benchmark** We performed the benchmark tasks from Lueckmann et al. [44] with NRE-C. Our architecture and hyperparameters deviate based on the results of the last paragraph. We applied the largest number of computationally practical contrastive parameters, namely $K = 99$, and set $\gamma = 1$. Since we identified potential architecture bottlenecks in Sections 3.1 and 3.3, we used Large NN for our results. The averaged results are visible in Table 1. Sequential estimates are prepended with an S, all other estimates are amortized. We trained a five amortized estimators per task with five seeds, then we predicted the posterior for each of the ten observations $\boldsymbol{x}_1, \ldots, \boldsymbol{x}_{10}$. Lueckmann et al. [44] trained an estimator for every $\boldsymbol{x}$ with a new seed. The reported C2ST for our method is averaged across ten posteriors per seed with five seeds, i.e., fifty C2ST computations per task. The other methods are averaged across one posterior per seed with ten seeds, i.e., ten C2ST computations per task.

NRE-C performed better than NRE-B at all budgets and generally well among the amortized algorithms. The high performance at $10^5$ simulation budget implies that our design has high capacity and may scale well. Sequential methods naturally lead to high exactness since they are tailored to the specific observation $\boldsymbol{x}_o$, but they come with drawbacks since they cannot practically perform empirical expected coverage tests [31, 48] nor bound the mutual information, as discussed in Section 2.2. Detailed results per task can be found in Appendix C.

## 4 Conclusion

We introduced a generalization of NRE-A and NRE-B which discourages the NRE-B specific $c_{\boldsymbol{w}}(\boldsymbol{x})$ bias term and produces an unbiased estimate of the likelihood-to-evidence ratio at optimum. This property has implications for the importance sampling diagnostic, which is critical to practitioners. We suggested using a variational bound on the mutual information as a model-selection metric that could replace the C2ST averaged over several observations. It is significantly more practical since it does not require access to the ground truth posterior; however, it remains a lower bound so we don't know the overall quality of our estimator. Further discussions and derivations related to the mutual information are available in Appendix D.

In the context of NRE-C, we found that increasing the number of contrastive parameters $K$ improved the C2ST in most cases, at the price of increasing the normalizing constant. Setting $\gamma \approx 1$ was generally the fastest to saturate and lower

Table 1: Averaged C2ST for various budgets on the simulation-based inference benchmark [44]. An S implies a sequential i.e. non-amortized method.

| | | | C2ST |
| Simulation budget
Algorithm | $10^3$ | $10^4$ | $10^5$ |
| --- | --- | --- | --- |
| NRE-C (ours) | 0.853 | 0.762 | 0.680 |
| REJ-ABC | 0.965 | 0.920 | 0.871 |
| NLE | 0.826 | 0.753 | 0.723 |
| NPE | 0.838 | 0.736 | 0.654 |
| NRE (NRE-B) | 0.867 | 0.811 | 0.762 |
| SMC-ABC | 0.948 | 0.873 | 0.816 |
| SNLE | 0.783 | 0.704 | 0.655 |
| SNPE | 0.796 | 0.677 | 0.615 |
| SNRE (SNRE-B) | 0.788 | 0.703 | 0.610 |

values seemed to slightly improve performance with the Large NN and fixed simulation data. These results indicate that better performance can be achieved by using NRE-C with these hyperparameters than the other version of NRE. When both training parameters $\boldsymbol{\theta}$ and simulation data $\boldsymbol{x}$ are unlimited, the architecture size plays an important role in the quality of surrogate model. We tried drawing contrastive parameters from the prior, a commonly available practical use-case, and found that it damped the effect of $K$ on the C2ST. According to the C2ST, for highest accuracy with a fixed budget set $K$ large, up to the mini-batch size divided by 2, and $\gamma \approx 1.0$ for the best convergence rate. When the prior is sampled for every mini-batch, applying a higher $K$ may be less valuable. In situations where the normalizing constant should be very close to unity, i.e., for practitioners who want to run diagnostics, drawing $\boldsymbol{\theta}$ from the prior with every mini-batch and using a smaller $K$ and $\gamma$ is best.

**Broader Impact**  The societal implications of SBI are similar to other inference methods. The methods are primarily scientific and generally lead to interesting discoveries; however, one must be careful to use an accurate generative model and to carefully test empirical results. Model mismatch and untested inference can lead to incorrect conclusions. That is why we emphasize the importance of the diagnostics in our paper. This nuance can be missed by practitioners doing inference in any field; however, special care should be taken when producing inference results that may be used for making decisions in areas like predicting hidden variables that describe human behavior or determining what factors are responsible for climate change. This list is non-comprehensive and not specific to SBI.

## Acknowledgments and Disclosure of Funding

We thank Noemi Anau Montel, James Alvey, Kosio Karchev for reading; Teodora Pandeva for mathematics consultation; Maud Canisius for design consultation; Marco Federici and Gilles Louppe for discussions. This work uses `numpy` [26], `scipy` [72], `seaborn` [73], `matplotlib` [32], `pandas` [52, 74], `pytorch` [57], and `jupyter` [38].

We want to thank the DAS-5 computing cluster for access to their TitanX GPUs. DAS-5 is funded by the NWO/NCF (the Netherlands Organization for Scientific Research). We thank SURF (www.surf.nl) for the support in using the Lisa Compute Cluster.

Benjamin Kurt Miller is funded by the University of Amsterdam Faculty of Science (FNWI), Informatics Institute (IvI), and the Institute of Physics (IoP). We received funding from the European Research Council (ERC) under the European Union's Horizon 2020 research and innovation programme (Grant agreement No. 864035 – UnDark).

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
