# A Relationship to other SBI methods

**Sampling in SBI**  All NRE (and SBI) algorithms require samples from the joint distribution $p(\boldsymbol{\theta}, \boldsymbol{x})$. NRE additionally requires samples from the product of marginals $p(\boldsymbol{\theta})p(\boldsymbol{x})$. Sampling the joint with a simulator requires drawing from the prior $\boldsymbol{\theta} \sim p(\boldsymbol{\theta})$ then passing that parameter into the simulator to produce $\boldsymbol{x} \sim p(\boldsymbol{x} \,|\, \boldsymbol{\theta})$. Sampling the product of marginals is simple, just take another sample from the prior $\boldsymbol{\theta}' \sim p(\boldsymbol{\theta})$ and pair it with our simulation from before. Then we have $(\boldsymbol{\theta}, \boldsymbol{x}) \sim p(\boldsymbol{\theta}, \boldsymbol{x})$ and $(\boldsymbol{\theta}', \boldsymbol{x}) \sim p(\boldsymbol{\theta})p(\boldsymbol{x})$. In practice, we refer to this operation as a bootstrap within a mini-batch where we take $\boldsymbol{\theta}'$ from other parameter-simulation pairs and reuse them to create samples drawn from the product of marginals $p(\boldsymbol{\theta})p(\boldsymbol{x})$. NRE-C sometimes requires more $\boldsymbol{\theta}$ samples, often represented as $\boldsymbol{\Theta}$. These can be generated by merely concatenating several samples from $p(\boldsymbol{\theta})$. Some SBI methods, e.g., sequential methods, sometimes replace the prior with a proposal distribution $\tilde{p}(\boldsymbol{\theta})$.

**Amortized and sequential SBI**  Recently, significant progress has been made in SBI, especially with so-called *sequential* methods that use active learning to draw samples from the posterior for a fixed observation-of-interest $\boldsymbol{x}_o$ [16, 18, 20, 21, 24, 35, 42, 53, 56, 59, 60, 61, 63, 69]. *Amortized* SBI algorithms, that can draw samples from the posterior for arbitrary observation $\boldsymbol{x}$, have also enjoyed attentive development [8, 9, 30, 36, 47, 48]. Hermans et al. [31] and Miller et al. [48] argue that their intrinsic empirical testability makes amortized methods better applicable to the scientific use-case despite their inherently higher training expense. The last pillar of development has been into assessment methods that determine the reliability of approximate inference results. In the machine learning community, the focus has been on evaluating the exactness of estimates for tractable problems [44]. Evaluation methods which apply in the practical case where no tractable posterior is available are under development [7, 12, 14, 29, 67]. We make a contribution to this effort in Appendix D.

**Contrastive learning, NRE-B, and NPE**  We call our method Contrastive Neural Ratio Estimation because it the classifier is trained to identify which pairs of $\boldsymbol{\theta}$ and $\boldsymbol{x}$ should be paired together. Gutmann et al. [25] created an overview of contrastive learning for statistical problems generally. Specific connections are in the loss functions with NRE-A closely corresponding to noise-contrastive estimation (NCE) [22, 23] and NRE-B with RankingNCE, InfoNCE, and related [45, 46, 49, 71].

A core aspect of our paper focus on the effects of the NRE-B ratio estimate biased by $c_{\boldsymbol{w}}(\boldsymbol{x})$ at optimum. Ma and Collins [45] also investigate the effects on the partition function when applying a binary or multi-class loss variant for estimating conditional energy-based models. Due to the similarity in the loss functions, they exhibit a similar bias in their partition function. In order to correct this bias they estimate $c_{\boldsymbol{w}}(\boldsymbol{x})$ directly. We did not attempt to do this for our problem, so we do not have intuition about the effectiveness of such an approach to the likelihood-based diagnostic in NRE. Although, since $c_{\boldsymbol{w}}(\boldsymbol{x})$ is completely unconstrained, it may be quite difficult to estimate. We believe that this would be an alternative direction for future work.

Durkan et al. [16] emphasize the connection between NRE and contrastive learning in their paper which created a framework such that Neural Posterior Estimation (NPE) and NRE-B can be trained using the same loss function. The addition of an independently drawn $(\boldsymbol{\Theta}, \boldsymbol{x})$ set means that the NRE-C framework is not trivially applicable for computing the normalizing constant on atoms necessary for their sequential version of NPE.

**Software**  Our experiments used `sbi` [68], which is released under an AGPL-3.0 license, and `sbibm` [44], which is released under an MIT license. They implement various neural SBI algorithms and benchmark problems respectively.

## A.1 Full derivation of other NRE methods using our framework

Here we present a derivation of the previous works NRE-A [30] and NRE-B [30] in our framwork.

**NRE-A** To estimate $r(\boldsymbol{x} \mid \boldsymbol{\theta})$, Hermans et al. [30] introduce an indicator variable $y$ which switches between dependently and independently drawn samples. We have conditional probability

$$p_{\text{NRE-A}}(\boldsymbol{\theta}, \boldsymbol{x} \mid y = k) = \begin{cases} p(\boldsymbol{\theta})p(\boldsymbol{x}) & k = 0 \\ p(\boldsymbol{\theta}, \boldsymbol{x}) & k = 1 \end{cases}. \tag{17}$$

Each class' marginal probability is set equally, $p(y = 0) = p(y = 1)$. Dropping the NRE-A subscript, the probability that $(\boldsymbol{\theta}, \boldsymbol{x})$ was drawn jointly is encoded in the another conditional probability

$$\begin{aligned} p(y = 1 \mid \boldsymbol{\theta}, \boldsymbol{x}) &= \frac{p(\boldsymbol{\theta}, \boldsymbol{x} \mid y = 1)p(y = 1)}{p(\boldsymbol{\theta}, \boldsymbol{x} \mid y = 1)p(y = 1) + p(\boldsymbol{\theta}, \boldsymbol{x} \mid y = 0)p(y = 0)} \\ &= \frac{p(\boldsymbol{\theta}, \boldsymbol{x})}{p(\boldsymbol{\theta}, \boldsymbol{x}) + p(\boldsymbol{\theta})p(\boldsymbol{x})} = \frac{r(\boldsymbol{x} \mid \boldsymbol{\theta})}{1 + r(\boldsymbol{x} \mid \boldsymbol{\theta})}. \end{aligned} \tag{18}$$

NRE-A estimates $\log \hat{r}(\boldsymbol{x} \mid \boldsymbol{\theta})$ with neural network $f_{\boldsymbol{w}}$ with weights $\boldsymbol{w}$. Training is done by minimizing the binary cross-entropy of $q_{\boldsymbol{w}}(y = 1 \mid \boldsymbol{\theta}, \boldsymbol{x}) \coloneqq \sigma \circ f_{\boldsymbol{w}}(\boldsymbol{\theta}, \boldsymbol{x})$ relative to $p(y, \boldsymbol{\theta}, \boldsymbol{x})$. For $B$ samples,

$$\boldsymbol{w} = \arg\min_{\boldsymbol{w}} \left[ -\frac{1}{B} \sum_{b=1}^{B} \log \left( 1 - \sigma \circ f_{\boldsymbol{w}}(\boldsymbol{\theta}^{(b)}, \boldsymbol{x}^{(b)}) \right) - \frac{1}{B} \sum_{b'=1}^{B} \log \left( \sigma \circ f_{\boldsymbol{w}}(\boldsymbol{\theta}^{(b')}, \boldsymbol{x}^{(b')}) \right) \right] \tag{19}$$

where $\boldsymbol{\theta}^{(b)}, \boldsymbol{x}^{(b)} \sim p(\boldsymbol{\theta})p(\boldsymbol{x})$ and $\boldsymbol{\theta}^{(b')}, \boldsymbol{x}^{(b')} \sim p(\boldsymbol{\theta}, \boldsymbol{x})$. In practice, $\boldsymbol{\theta}^{(b')}$ is bootstrapped from within the mini-batch to produce $\boldsymbol{\theta}^{(b)}$. NRE-A's ratio estimate converges to $f_{\boldsymbol{w}} = \log \frac{p(\boldsymbol{x} \mid \boldsymbol{\theta})}{p(\boldsymbol{x})} = \log r(\boldsymbol{x} \mid \boldsymbol{\theta})$ given unlimited model flexibility and data

**NRE-B** Durkan et al. [16] estimate $r(\boldsymbol{x} \mid \boldsymbol{\theta})$ by training a classifier that selects from among $K$ parameters $\boldsymbol{\Theta} \coloneqq (\boldsymbol{\theta}_1, \ldots, \boldsymbol{\theta}_K)$ which could have generated $\boldsymbol{x}$. In contrast with NRE-A, one of these parameters $\boldsymbol{\theta}_k$ is *always* drawn jointly with $\boldsymbol{x}$. Let $y$ be a random variable which indicates which one of $K$ parameters simulated $\boldsymbol{x}$. The marginal probability $p(y = k) \coloneqq 1/K$ is uniform. That means

$$p_{\text{NRE-B}}(\boldsymbol{\Theta}, \boldsymbol{x} \mid y = k) \coloneqq p(\boldsymbol{\theta}_1) \cdots p(\boldsymbol{\theta}_K)p(\boldsymbol{x} \mid \boldsymbol{\theta}_k) \tag{20}$$

defines our conditional probability for parameters and data. Bayes' rule reveals a conditional distribution over $y$, dropping the NRE-B subscript, therefore

$$\begin{aligned} p(y = k \mid \boldsymbol{\Theta}, \boldsymbol{x}) &= \frac{p(\boldsymbol{\Theta}, \boldsymbol{x} \mid y = k)p(y = k)}{p(\boldsymbol{\Theta}, \boldsymbol{x})} = \frac{p(\boldsymbol{\Theta}, \boldsymbol{x} \mid y = k)p(y = k)}{\sum_i p(\boldsymbol{\Theta}, \boldsymbol{x} \mid y = i)p(y = i)} \\ &= \frac{p(\boldsymbol{\theta}_1) \cdots p(\boldsymbol{\theta}_k, \boldsymbol{x}) \cdots p(\boldsymbol{\theta}_K)}{\sum_i p(\boldsymbol{\theta}_1) \cdots p(\boldsymbol{\theta}_i, \boldsymbol{x}) \cdots p(\boldsymbol{\theta}_K)} = \frac{p(\boldsymbol{\theta}_k \mid \boldsymbol{x})/p(\boldsymbol{\theta}_k)}{\sum_i p(\boldsymbol{\theta}_i \mid \boldsymbol{x})/p(\boldsymbol{\theta}_i)} = \frac{r(\boldsymbol{x} \mid \boldsymbol{\theta}_k)}{\sum_i r(\boldsymbol{x} \mid \boldsymbol{\theta}_i)}. \end{aligned} \tag{21}$$

NRE-B estimates $\log \hat{r}(\boldsymbol{x} \mid \boldsymbol{\theta})$ with a neural network $g_{\boldsymbol{w}}$. Training is done by minimizing the cross entropy of $q_{\boldsymbol{w}}(y = k \mid \boldsymbol{\Theta}, \boldsymbol{x}) \coloneqq \exp \circ g_{\boldsymbol{w}}(\boldsymbol{\theta}_k, \boldsymbol{x}) / \sum_i \exp \circ g_{\boldsymbol{w}}(\boldsymbol{\theta}_i, \boldsymbol{x})$ relative to $p(y, \boldsymbol{x}, \boldsymbol{\theta})$;

$$\boldsymbol{w} = \arg\min_{\boldsymbol{w}} \mathbb{E}_{p(y, \boldsymbol{\Theta}, \boldsymbol{x})} \left[ -q_{\boldsymbol{w}}(y \mid \boldsymbol{\Theta}, \boldsymbol{x}) \right] \approx \arg\min_{\boldsymbol{w}} \left[ -\frac{1}{B} \sum_{b'=1}^{B} \log \frac{\exp \circ g_{\boldsymbol{w}}(\boldsymbol{\theta}_k^{(b')}, \boldsymbol{x}^{(b')})}{\sum_i \exp \circ g_{\boldsymbol{w}}(\boldsymbol{\theta}_i^{(b')}, \boldsymbol{x}^{(b')})} \right] \tag{22}$$

where $\boldsymbol{\theta}_1^{(b')}, \ldots, \boldsymbol{\theta}_K^{(b')} \sim p(\boldsymbol{\theta})$, $k^{(b')} \sim p(y)$, and $\boldsymbol{x}^{(b')} \sim p(\boldsymbol{x} \mid \boldsymbol{\theta}_k^{(b')})$ over $B$ samples. In our parameterization and given unlimited flexibility and data, $g_{\boldsymbol{w}}(\boldsymbol{\theta}, \boldsymbol{x}) = \log \frac{p(\boldsymbol{\theta} \mid \boldsymbol{x})}{p(\boldsymbol{\theta})} + c_{\boldsymbol{w}}(\boldsymbol{x})$ at convergence. The extra term enters because the optimal classifier for (22) need not be normalized.

# B Theoretical Arguments

We present first the proof of convergence for NRE-C. Afterwards, we discuss the properties of estimated importance weights in NRE-B.

## B.1 Proof of convergence of NRE-C

**Lemma 1.** *Consider for $k = 0, \ldots, K$ the following probability distributions for $\boldsymbol{z}$:*

$$p(\boldsymbol{z} \mid y = k). \tag{23}$$

*and $p(y) > 0$ a probability distribution for $y$. Put $p_k := p(y = k)$ for $k = 1, \ldots, K$. For functions $f_k : \mathcal{Z} \to \mathbb{R}$, $k = 1, \ldots, K$, let:*

$$q(y = k \mid f, \boldsymbol{z}) := \begin{cases} \frac{1}{1 + \sum_{j=1}^{K} \frac{p_j}{p_0} \exp(f_j(\boldsymbol{z}))}, & k = 0, \\ \frac{\frac{p_k}{p_0} \exp(f_k(\boldsymbol{z}))}{1 + \sum_{j=1}^{K} \frac{p_j}{p_0} \exp(f_j(\boldsymbol{z}))}, & k = 1, \ldots, K. \end{cases} \tag{24}$$

*Note that $q(y = k \mid f, \boldsymbol{z}) > 0$ for all $k = 0, \ldots, K$ and that $\sum_{k=0}^{K} q(y = k \mid f, \boldsymbol{z}) = 1$ for every $K$-tuple $f = (f_k)_{k=1,\ldots,K}$ and $\boldsymbol{z} \in \mathcal{Z}$. Consider a minimizer:*

$$f^* \in \arg\min_f \mathbb{E}_{p(\boldsymbol{z} \mid y)p(y)} \left[ -\log q(y \mid f, \boldsymbol{z}) \right]. \tag{25}$$

*Then we have for $p(\boldsymbol{z})$-almost-all $\boldsymbol{z}$ and all $k = 1, \ldots, K$:*

$$f_k^*(\boldsymbol{z}) = \log \frac{p(\boldsymbol{z}|y = k)}{p(\boldsymbol{z}|y = 0)}. \tag{26}$$

*Proof.* We have:

$$f^* \in \arg\min_f \mathbb{E}_{p(\boldsymbol{z} \mid y)p(y)} \left[ -\log q(y \mid f, \boldsymbol{z}) \right] \tag{27}$$

$$= \arg\min_f \mathbb{E}_{p(\boldsymbol{z},y)} \left[ \log \frac{p(y \mid \boldsymbol{z})}{q(y \mid f, \boldsymbol{z})} \right] \tag{28}$$

$$= \arg\min_f \mathbb{E}_{p(\boldsymbol{z})} \left[ \text{KLD}(p(y \mid \boldsymbol{z}) \| q(y \mid f, \boldsymbol{z})) \right], \tag{29}$$

which is minimized, when KLD $= 0$, thus:

$$0 = \text{KLD}(p(y \mid \boldsymbol{z}) \| q(y \mid f^*, \boldsymbol{z})), \tag{30}$$

which implies that for $p(\boldsymbol{z})$-almost-all $\boldsymbol{z}$:

$$q(y \mid f^*, \boldsymbol{z}) = p(y \mid \boldsymbol{z}) = \frac{p(\boldsymbol{z}|y)}{p(\boldsymbol{z})} p(y). \tag{31}$$

So we get with the definition of $q(y \mid f^*, \boldsymbol{z})$:

$$\frac{p(\boldsymbol{z}|y = 0)}{p(\boldsymbol{z})} p_0 = \frac{1}{1 + \sum_{j=1}^{K} \frac{p_j}{p_0} \exp(f_j^*(\boldsymbol{z}))}, \qquad k = 0, \tag{32}$$

$$\frac{p(\boldsymbol{z}|y = k)}{p(\boldsymbol{z})} p_k = \frac{\frac{p_k}{p_0} \exp(f_k^*(\boldsymbol{z}))}{1 + \sum_{j=1}^{K} \frac{p_j}{p_0} \exp(f_j^*(\boldsymbol{z}))}, \qquad k = 1, \ldots, K. \tag{33}$$

Dividing the latter by the former gives for $k = 1, \ldots, K$:

$$\frac{p(\boldsymbol{z}|y = k)}{p(\boldsymbol{z}|y = 0)} \frac{p_k}{p_0} = \frac{p_k}{p_0} \exp(f_k^*(\boldsymbol{z})), \tag{34}$$

implying for $k = 1, \ldots, K$ and $p(\boldsymbol{z})$-almost-all $\boldsymbol{z}$:

$$f_k^*(\boldsymbol{z}) = \log \frac{p(\boldsymbol{z}|y = k)}{p(\boldsymbol{z}|y = 0)}. \tag{35}$$

This shows the claim. $\qquad\square$

We used the symbol KLD to imply the Kullback-Leibler Divergence. The proof in Lemma 1 is slightly more general than necessary for our typical case. All our functions $f_k$ are typically the same, namely the evaluation of a neural network with weights $\boldsymbol{w}$. Rather than searching for the function $f$ which minimizes the objective, we search for the weights, but these are equivalent. Finally to make everything fit, we set $\boldsymbol{z} := (\boldsymbol{\theta}, \boldsymbol{x})$.

## B.2 Properties of the importance sampling diagnostic on biased ratio estimates

In Section 2.2 we discuss the importance sampling diagnostic, in which the estimated ratio is tested by comparing weighted samples from $\boldsymbol{x} \sim p(\boldsymbol{x})$ to unweighted samples from $\boldsymbol{x} \sim p(\boldsymbol{x} \mid \boldsymbol{\theta})$. The estimated ratio from NRE-C is merely $\exp(h_{\boldsymbol{w}}(\boldsymbol{\theta}, \boldsymbol{x}))$ and is therefore not restricted to have the properties of a ratio of probability distributions, except at optimum with unlimited flexibility and data. The importance sampling diagnostic is designed to test whether the estimated ratio is close to having these properties for a fixed $\boldsymbol{\theta}$. One way to improve ratios estimated by NRE-C is to compute the normalizing constant $Z_{\boldsymbol{w}}(\boldsymbol{x})$, which should be close to one, and replace the ratio with this "normalized" version.

The unrestricted nature of the NRE-B-specific $c_{\boldsymbol{w}}(\boldsymbol{x})$ bias means that the normalization constant $Z_{\boldsymbol{w}}(\boldsymbol{x})$ does not have to be close to one. Further, we show that the unrestricted bias means that the normalizing constant for two NRE-B ratio estimators, which are equivalent in terms of their loss function (22), is not unique, i.e., the estimate is ill-posed. This property means that normalizing NRE-B will not, in general, produce an estimator which passes the diagnostic.

Consider two ratio estimates with the relationship $\hat{r}_1(\boldsymbol{x} \mid \boldsymbol{\theta}) = \hat{r}_2(\boldsymbol{x} \mid \boldsymbol{\theta})/C(\boldsymbol{x})$ where $C$ is an positive function of $\boldsymbol{x}$ resulting from the aggregation of the exponentiated bias. Given $N$ samples $\boldsymbol{x}_n \sim p(\boldsymbol{x})$ with weights $w_1(\boldsymbol{x}_n) = \hat{r}_1(\boldsymbol{x}_n \mid \boldsymbol{\theta})$ and $w_2(\boldsymbol{x}_n) = \hat{r}_2(\boldsymbol{x}_n \mid \boldsymbol{\theta})$, we can compute the importance-normalized weights by

$$\bar{w}_1(\boldsymbol{x}_n) = \frac{w_1(\boldsymbol{x}_n)}{\sum_{i=1}^{N} w_1(\boldsymbol{x}_i)}, \qquad \bar{w}_2(\boldsymbol{x}_n) = \frac{w_2(\boldsymbol{x}_n)}{\sum_{i=1}^{N} w_2(\boldsymbol{x}_i)}. \tag{36}$$

However, if we substitute this constant back into our expression, we find that the weights do not agree

$$\bar{w}_1(\boldsymbol{x}_n) = \frac{w_2(\boldsymbol{x}_n)/C(\boldsymbol{x}_n)}{\sum_{i=1}^{N} w_2(\boldsymbol{x}_i)/C(\boldsymbol{x}_i)} \neq \bar{w}_2(\boldsymbol{x}_n). \tag{37}$$

Therefore, normalization does not "protect" against scaling bias introduced by a function in $\boldsymbol{x}$. NRE-B does not penalize functional biases like $C(\boldsymbol{x})$, so even ratios considered optimal by NRE-B can easily fail the diagnostic. Meanwhile, NRE-C encourages terms like $C(\boldsymbol{x})$ towards one. That has the effect of making the $C(\boldsymbol{x}) \approx 1$ drop out of the normalized importance weights, thus making performance on the diagnostic indicative of a better ratio estimate, given enough classifier flexibility.

## C  Experimental Details

**Computational costs**  Our experiments were performed on a cluster of Nvidia Titan V graphics processing units. The primary expense was the hyperparameter search within Sections 3.1, 3.2, and the first part of Section 3.3. The run of those experiments took about 50,000 gpu-minutes total considering all of our current data. Since there were several iterations, we multiply this by three to estimate total compute. The next expense was the computation of sbibm in Section 3.3 that took about 24,000 gpu-minutes. Together these equal about 2880 gpu-hours. An estimate for the total carbon contributions corresponds to 311.04 kgCO$_2$. Luckily, our clusters run completely on wind power offsetting the contribution. Estimations were conducted using the MachineLearning Impact calculator presented in [39].

**Architecture and training**  The centerpiece of our method is the neural network $h_{\boldsymbol{w}}$ which is trained as a classifier. The hyperparameter choices here were fairly constant throughout the experiments. Any hyperparameters for training or architecture which were consistent across experiments are listed

in Table 2. Hidden features and number of RESNET blocks depend on the architecture. Large NN uses three resnet blocks with 128 features, while Small NN uses two resnet blocks and 50 features.

Table 2: Architecture

| Hyperparameter | Value |
| --- | --- |
| Activation Function | RELU |
| AMSGRAD | No |
| Architecture | RESNET |
| Batch normalization | Yes |
| Batch size | 1024 |
| Dropout | No |
| Max epochs | 1000 |
| Learning rate | 0.0005 |
| Optimizer | ADAM |
| Weight Decay | 0.0 |
| Standard-score Observations | Using first batch |
| Standard-score Parameters | Using first batch |

## C.1 Hyperparameter search measured with C2ST

In this section, we trained many neural networks with different hyperparameter settings on three different tasks from Lueckmann et al. [44]. We trained both architectures, Large and Small NN, on a grid of $\gamma$ and $K$ values with NRE-B and NRE-C. No matter whether we were training with unlimited draws from the joint, prior, or fixed data we designed an epoch such that it has 20 training mini-batches and 2 validation mini-batches. For fixed initial data and prior this corresponds to a simulation budget of 22,528. The mean validation losses per-epoch for these networks is visible in Figures 7, 8, and 9.

For one specific problem, we also computed validation loss at a fixed $\gamma$ and $K$ no matter the training regime in an effort to produce comparable validation losses, specifically $\gamma = 1$ and $K = 1$ aka the NRE-A regime. This training setting reverses some of the trends we've seen when validated on the same loss as training data. We did not apply it further since this could bias results. See Figure 11. To compare, we also plot the validation loss on the sample problem but using the $\gamma$ and $K$ each model was trained with. See Figure 10, note that the bias depending on $\gamma$ and $K$ is clearly visible (just as in the other validation loss plots).

Once the networks were trained, we drew samples from ten per-task posteriors based on ten predefined observations $x_i$ with $i \in 1, 2, \ldots, 10$. This leveraged the amortized property of NRE-C. Samples were drawn on these problems using rejection sampling. Once the samples were drawn, they were compared to ground truth posterior samples from the benchmark with the C2ST. The detailed plot which shows per-task behavior is available in Figure 6.

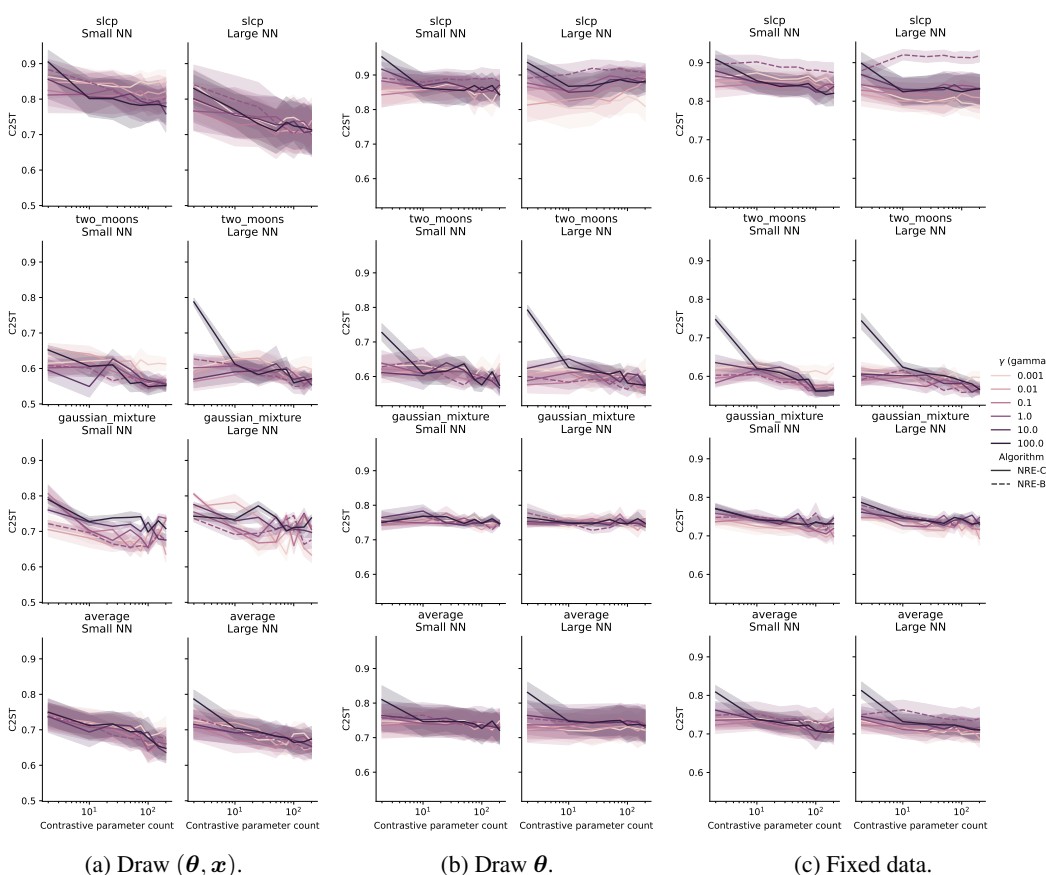

(a) Draw $(\boldsymbol{\theta}, \boldsymbol{x})$.  (b) Draw $\boldsymbol{\theta}$.  (c) Fixed data.

Figure 6: A measure of exactness comparing to the ground truth to the samples from a surrogate posterior, the C2ST, is plotted as a function of number of contrastive parameters. Each row corresponds to a different task: SLCP, Two Moons, Gaussian Mixture, and an average of the results across tasks. Both NRE-C and NRE-B are shown along with various $\gamma$ values, and architectures are shown. Recall that C2ST assigns 1.0 to inaccurate and 0.5 to accurate approximations. (a) Corresponds to Section 3.1 where unlimited draws from the joint are allowed. (b) Corresponds to Section 3.2 where unlimited draws from the prior are allowed but the $\boldsymbol{x}$ data is fixed. (c) Corresponds to Section 3.3 where both the initial draws of $\boldsymbol{\theta}$ and $\boldsymbol{x}$ are the only data available.

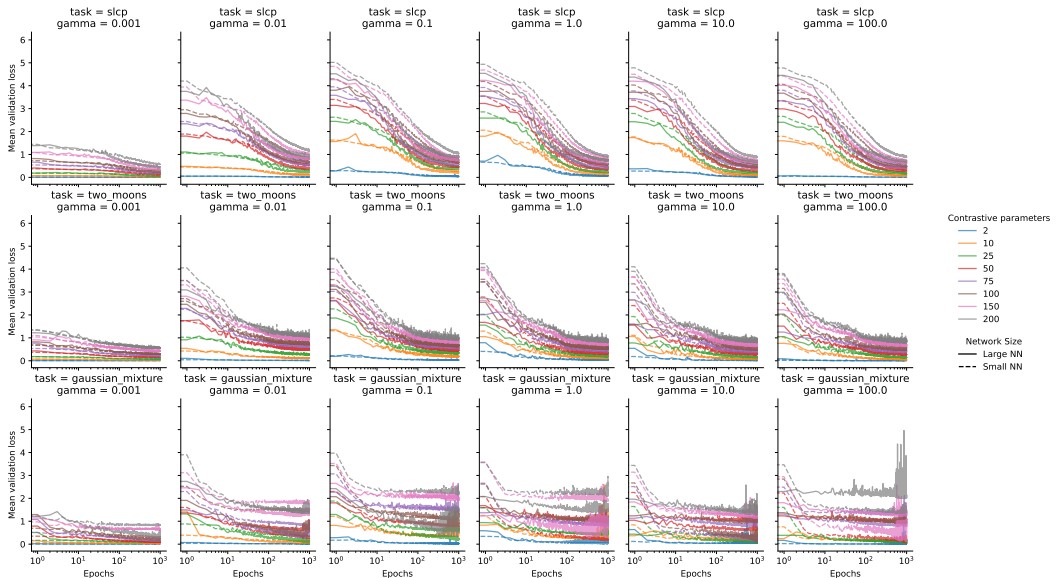

Figure 7: The validation loss from NRE-C is reported versus epochs on various tasks, $\gamma$, $K$, and architectures trained using unlimited draws from the joint distribution. The rows correspond to different tasks, columns to different $\gamma$, colors to different $K$, and dashed or solid lines to Small and Large NN respectively. These plots correspond with the technique discussed in Section 3.1.

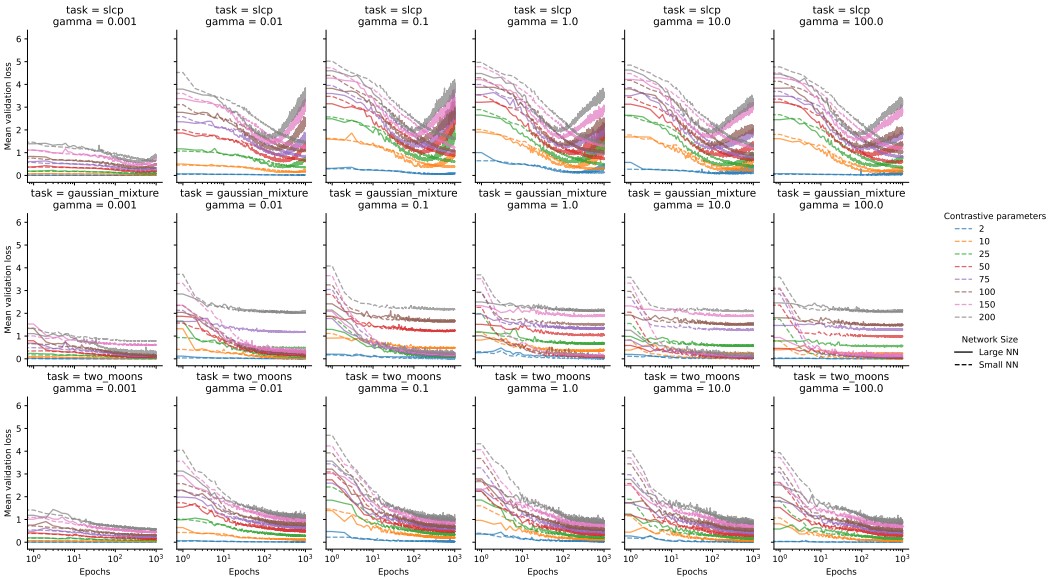

Figure 8: The validation loss from NRE-C is reported versus epochs on various tasks, $\gamma$, $K$, and architectures trained using a simulation budget of 22,528 but unlimited draws from the prior during training. The rows correspond to different tasks, columns to different $\gamma$, colors to different $K$, and dashed or solid lines to Small and Large NN respectively. These plots correspond with the technique discussed in Section 3.2.

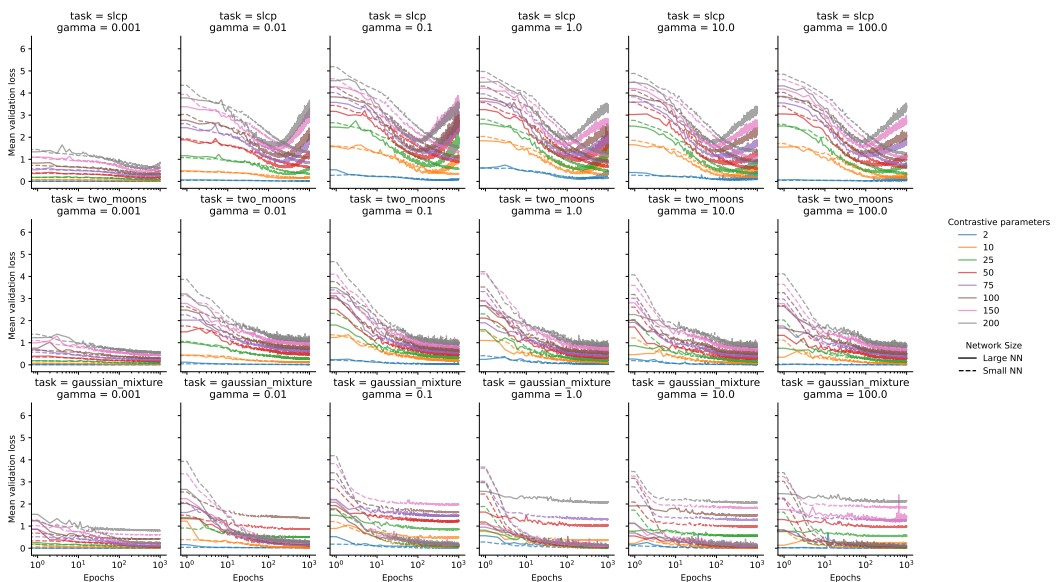

Figure 9: The validation loss from NRE-C is reported versus epochs on various tasks, $\gamma$, $K$, and architectures trained using a fixed simulation budget of 22,528. The rows correspond to different tasks, columns to different $\gamma$, colors to different $K$, and dashed or solid lines to Small and Large NN respectively. These plots correspond with the technique discussed in Section 3.3.

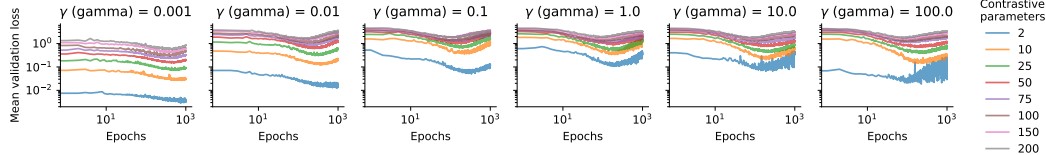

Figure 10: The validation loss from NRE-C is reported versus epochs on SLCP with the Large NN where colors indicate different contrastive parameter counts. The plot shows the convergence rates of each model. Since validation loss is a function of $K$ and $\gamma$, the relative performance of models is *not* comparable. A grid search of $K$ and $\gamma$ indicates that increasing $K$ leads to earlier convergence at fixed $\gamma$. With $K$ fixed, $\gamma < 1$ has a negative effect on convergence rate and $\gamma > 1$ is ambiguous.

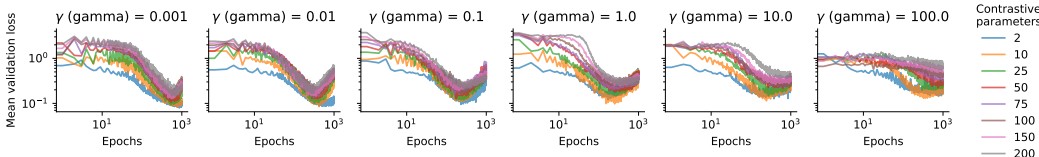

Figure 11: The validation loss from NRE-C is reported versus epochs on SLCP with the Large NN where colors indicate different contrastive parameter counts, i.e, $K + 1$. A fixed validation loss was used, namely $\ell_{1,1}$. Although the validation loss is now comparable so that we can see the performance of the different classifiers on the same task, some of the convergence rate trends disagreed with SLCP in Figure 10. Naturally, the classifier trained to distinguish two samples often performed best on this task.

## C.2 Simulation-based inference benchmark

We trained NRE-C with $\gamma = 1.0$ and $K = 100$ using Large NN for 1000 epochs on all of the tasks from the simulation-based inference benchmark [44]. In contrast with our previous hyperparameter search experiments, we used their Markov-chain Monte Carlo sampling scheme which applies slice sampling [51]. The details results are presented in Table 3 and Table 4. There were three tasks which did not succeed with this sampling procedure, there we used rejection sampling.

Our training slightly diverged from the benchmark. We enumerate the ways below.

- Large NN is a bigger architecture than the one which produced their reported NRE-B values. In fact, the NRE-B version has the same settings as Small NN; however, we found that larger networks performed better in our search and chose the bigger one for that reason.

- In the benchmark, networks are trained with early stopping but we always trained for 1000 epochs. We selected the network which performed best on the validation loss, using the same values for $K$ and $\gamma$ in the validation loss as in the training.

- Even though NRE-B is an amortized method, the benchmark trained a network for every observation. Instead, we leveraged the amortization properties of NRE-C and trained a single network which drew samples from each approximate posterior given by each of the ten observations.

**Task details**  We provide a short summary of all of the inference tasks in the SBI benchmark by Lueckmann et al. [44].

**Bernoulli GLM**  This task is a generalized linear model. The likelihood is Bernoulli distributed. The data is a 10-dimensional summary statistic from an 100-dimensional raw vector. The posterior is 10-dimensional and it only has one mode.

**Bernoulli GLM Raw**  This is the same task as above, but instead the entire 100-dimensional observation is shown to the inference method rather than the summary statistic.

**Gaussian Linear**  A simple task with a Gaussian distributed prior and a Gaussian likelihood over the mean. Both have a $\Sigma = 0.1 \cdot I$ covariance matrix. The posterior is also Gaussian. It is performed in 10-dimensions for the observations and parameters.

**Gaussian Linear Uniform**  This is the same as the task above, but instead the prior over the mean is a 10-dimensional uniform distribution from -1 to 1 in every dimension.

**Gaussian Mixture**  This task occurs in the ABC literature often. Infer the common mean of a mixture of Gaussians where one has covariance matrix $\Sigma = 1.0 \cdot I$ and the other $\Sigma = 0.01 \cdot I$. It occurs in two dimensions.

**Lotka Volterra**  This is an ecological predator-prey model where the simulations are generated from randomly drawn initial conditions by solving a parameterized differential equation. There are four parameters that control the coupling between the generation and destruction of both prey and predators. The priors are log normal. The data is a twenty dimensional summary statistic.

**SIR**  An epidemiological model simulating the progress of an contagious disease outbreak through a population. Simulations are generated from randomly drawn initial conditions with a parameterized differential equation defining the dynamics. There are two parameters with a log normal prior. The data is a ten dimensional summary statistic.

**SLCP**  A task which has a very simple non-spherical Gaussian likelihood, but a complex posterior over the five parameters which, via a non-linear function, define the mean and covariance of the likelihood. There are five parameters each with a uniform prior from -3 to 3. The data is four-dimensional but we take two samples from it. It was introduced in [56].

**SLCP with Distractors**  This is the same task as above but instead the data is concatenated with 92 dimensions of Gaussian noise.

**Two Moons**  This task exhibits a crescent shape posterior with bi-modality–two of the attributes often used to stump MCMC samplers. Both the data and parameters are two dimensional. The prior is uniform from -1 to 1.

Table 3: Simulation-based inference benchmark results.

| Task | Simulation budget Algorithm | $10^3$ | $10^4$ | C2ST $10^5$ |
|---|---|---|---|---|
| Bernoulli GLM | NRE-C (ours) | 0.829 | 0.688 | 0.617 |
| | REJ-ABC | 0.994 | 0.976 | 0.941 |
| | NLE | 0.740 | 0.605 | 0.545 |
| | NPE | 0.863 | 0.678 | 0.559 |
| | NRE (NRE-B) | 0.899 | 0.812 | 0.751 |
| | SMC-ABC | 0.991 | 0.981 | 0.818 |
| | SNLE | 0.634 | 0.553 | 0.522 |
| | SNPE | 0.855 | 0.614 | 0.525 |
| | SNRE (SNRE-B) | 0.718 | 0.584 | 0.529 |
| Bernoulli GLM Raw | NRE-C (ours) | 0.952 | 0.761 | 0.627 |
| | REJ-ABC | 0.995 | 0.984 | 0.966 |
| | NLE | 0.870 | 0.939 | 0.951 |
| | NPE | 0.900 | 0.765 | 0.607 |
| | NRE (NRE-B) | 0.915 | 0.834 | 0.777 |
| | SMC-ABC | 0.990 | 0.959 | 0.943 |
| | SNLE | 0.990 | 0.973 | 0.987 |
| | SNPE | 0.906 | 0.658 | 0.607 |
| | SNRE (SNRE-B) | 0.880 | 0.675 | 0.552 |
| Gaussian Linear | NRE-C (ours) | 0.684 | 0.583 | 0.547 |
| | REJ-ABC | 0.913 | 0.858 | 0.802 |
| | NLE | 0.650 | 0.555 | 0.515 |
| | NPE | 0.694 | 0.552 | 0.506 |
| | NRE (NRE-B) | 0.672 | 0.560 | 0.536 |
| | SMC-ABC | 0.922 | 0.829 | 0.726 |
| | SNLE | 0.628 | 0.548 | 0.519 |
| | SNPE | 0.652 | 0.544 | 0.507 |
| | SNRE (SNRE-B) | 0.670 | 0.536 | 0.515 |
| Gaussian Linear Uniform | NRE-C (ours) | 0.751 | 0.677 | 0.553 |
| | REJ-ABC | 0.977 | 0.948 | 0.909 |
| | NLE | 0.723 | 0.548 | 0.506 |
| | NPE | 0.696 | 0.553 | 0.509 |
| | NRE (NRE-B) | 0.788 | 0.706 | 0.631 |
| | SMC-ABC | 0.968 | 0.928 | 0.794 |
| | SNLE | 0.657 | 0.552 | 0.509 |
| | SNPE | 0.631 | 0.527 | 0.507 |
| | SNRE (SNRE-B) | 0.681 | 0.606 | 0.536 |
| Gaussian Mixture | NRE-C (ours) | 0.807 | 0.751 | 0.751 |
| | REJ-ABC | 0.883 | 0.789 | 0.772 |
| | NLE | 0.812 | 0.731 | 0.757 |
| | NPE | 0.731 | 0.661 | 0.555 |
| | NRE (NRE-B) | 0.784 | 0.752 | 0.734 |
| | SMC-ABC | 0.799 | 0.746 | 0.664 |
| | SNLE | 0.701 | 0.702 | 0.624 |
| | SNPE | 0.697 | 0.583 | 0.533 |
| | SNRE (SNRE-B) | 0.723 | 0.662 | 0.542 |

Table 4: Simulation-based inference benchmark results continued.

| Task | Simulation budget Algorithm | $10^3$ | $10^4$ | C2ST $10^5$ |
|------|------|------|------|------|
| | NRE-C (ours) | 1.000 | 0.977 | 0.983 |
| | REJ-ABC | 1.000 | 1.000 | 0.998 |
| | NLE | 0.994 | 0.956 | 0.952 |
| | NPE | 0.999 | 0.997 | 0.981 |
| Lotka-Volterra | NRE (NRE-B) | 1.000 | 0.998 | 0.996 |
| | SMC-ABC | 1.000 | 0.996 | 0.995 |
| | SNLE | 0.909 | 0.738 | 0.695 |
| | SNPE | 0.990 | 0.953 | 0.928 |
| | SNRE (SNRE-B) | 0.971 | 0.848 | 0.831 |
| | NRE-C (ours) | 0.780 | 0.673 | 0.578 |
| | REJ-ABC | 0.964 | 0.838 | 0.713 |
| | NLE | 0.761 | 0.748 | 0.730 |
| | NPE | 0.815 | 0.680 | 0.585 |
| SIR | NRE (NRE-B) | 0.841 | 0.770 | 0.690 |
| | SMC-ABC | 0.921 | 0.626 | 0.613 |
| | SNLE | 0.745 | 0.745 | 0.650 |
| | SNPE | 0.638 | 0.561 | 0.575 |
| | SNRE (SNRE-B) | 0.637 | 0.646 | 0.547 |
| | NRE-C (ours) | 0.973 | 0.941 | 0.810 |
| | REJ-ABC | 0.982 | 0.973 | 0.961 |
| | NLE | 0.946 | 0.771 | 0.699 |
| | NPE | 0.975 | 0.901 | 0.831 |
| SLCP | NRE (NRE-B) | 0.972 | 0.947 | 0.919 |
| | SMC-ABC | 0.982 | 0.969 | 0.963 |
| | SNLE | 0.921 | 0.713 | 0.578 |
| | SNPE | 0.965 | 0.845 | 0.666 |
| | SNRE (SNRE-B) | 0.968 | 0.917 | 0.721 |
| | NRE-C (ours) | 0.982 | 0.976 | 0.811 |
| | REJ-ABC | 0.988 | 0.987 | 0.987 |
| | NLE | 0.987 | 0.961 | 0.905 |
| | NPE | 0.982 | 0.970 | 0.863 |
| SLCP Distractors | NRE (NRE-B) | 0.980 | 0.968 | 0.953 |
| | SMC-ABC | 0.986 | 0.987 | 0.985 |
| | SNLE | 0.992 | 0.949 | 0.883 |
| | SNPE | 0.978 | 0.931 | 0.778 |
| | SNRE (SNRE-B) | 0.981 | 0.974 | 0.766 |
| | NRE-C (ours) | 0.777 | 0.594 | 0.526 |
| | REJ-ABC | 0.960 | 0.847 | 0.664 |
| | NLE | 0.773 | 0.713 | 0.668 |
| | NPE | 0.725 | 0.606 | 0.542 |
| Two Moons | NRE (NRE-B) | 0.822 | 0.761 | 0.629 |
| | SMC-ABC | 0.922 | 0.707 | 0.663 |
| | SNLE | 0.657 | 0.571 | 0.582 |
| | SNPE | 0.643 | 0.554 | 0.530 |
| | SNRE (SNRE-B) | 0.651 | 0.582 | 0.563 |

# D   Mutual Information

Estimating the mutual information is closely related to estimating the likelihood-to-evidence ratio. We reference various bounds on the mutual information and show how NRE-C can estimate them numerically. These bounds obey the variational principle and might be a practical candidate for validating the performance of SBI methods for scientific purposes, i.e., when the ground truth posterior is intractable. The approximation of the mutual information could synergize with other diagnostics like empirical, expected coverage testing and the importance sampling diagnostic. See section 2.2.

**Mutual information and expected Kullback-Leibler divergence**   If $p(\boldsymbol{\theta} \,|\, \boldsymbol{x})$ denotes the true posterior and $p_{\boldsymbol{w}}(\boldsymbol{\theta} \,|\, \boldsymbol{x})$ an approximate posterior then the quality of that approximation can be measured via the (forward) Kullback-Leibler divergence:

$$\mathrm{KLD}(p(\boldsymbol{\theta} \,|\, \boldsymbol{x}) \,\|\, p_{\boldsymbol{w}}(\boldsymbol{\theta} \,|\, \boldsymbol{x})). \tag{38}$$

However, in the SBI-setting we have only access to the likelihood $p(\boldsymbol{x} \,|\, \boldsymbol{\theta})$ via samples. Since we have also access to the prior $p(\boldsymbol{\theta})$ (analytically and) via samples we can sample from the joint $p(\boldsymbol{\theta}, \boldsymbol{x})$. So using the *expected Kullback-Leibler divergence* is more tractable in the SBI setting to measure the discrepancy between true and approximated prior:

$$\mathbb{E}_{p(\boldsymbol{x})}\left[\mathrm{KLD}(p(\boldsymbol{\theta} \,|\, \boldsymbol{x}) \,\|\, p_{\boldsymbol{w}}(\boldsymbol{\theta} \,|\, \boldsymbol{x}))\right]. \tag{39}$$

In all considered cases in this paper the approximate posterior $p_{\boldsymbol{w}}(\boldsymbol{\theta} \,|\, \boldsymbol{x})$ is given by:

$$p_{\boldsymbol{w}}(\boldsymbol{\theta} \,|\, \boldsymbol{x}) = \frac{\hat{r}_{\boldsymbol{w}}(\boldsymbol{x} \,|\, \boldsymbol{\theta})}{Z_{\boldsymbol{w}}(\boldsymbol{x})} p(\boldsymbol{\theta}), \qquad\qquad Z_{\boldsymbol{w}}(\boldsymbol{x}) := \int \hat{r}_{\boldsymbol{w}}(\boldsymbol{x} \,|\, \boldsymbol{\theta}) p(\boldsymbol{\theta}) \, d\boldsymbol{\theta}, \tag{40}$$

where $\hat{r}_{\boldsymbol{w}}(\boldsymbol{x} \,|\, \boldsymbol{\theta})$ comes from a (trained) neural network and $Z_{\boldsymbol{w}}(\boldsymbol{x})$ denotes the normalization constant. With the above notations we get for the expected Kullback-Leibler divergence the expression:

$$\mathbb{E}_{p(\boldsymbol{x})}\left[\mathrm{KLD}(p(\boldsymbol{\theta} \,|\, \boldsymbol{x}) \,\|\, p_{\boldsymbol{w}}(\boldsymbol{\theta} \,|\, \boldsymbol{x}))\right] = \mathbb{E}_{p(\boldsymbol{\theta}, \boldsymbol{x})}\left[\log \frac{p(\boldsymbol{\theta} \,|\, \boldsymbol{x})}{p_{\boldsymbol{w}}(\boldsymbol{\theta} \,|\, \boldsymbol{x})}\right] \tag{41}$$

$$= \mathbb{E}_{p(\boldsymbol{\theta}, \boldsymbol{x})}\left[\log \frac{p(\boldsymbol{\theta}, \boldsymbol{x})}{p(\boldsymbol{\theta}) p(\boldsymbol{x})} \frac{p(\boldsymbol{\theta})}{p_{\boldsymbol{w}}(\boldsymbol{\theta} \,|\, \boldsymbol{x})}\right] \tag{42}$$

$$= I(\boldsymbol{\theta}; \boldsymbol{x}) - \mathbb{E}_{p(\boldsymbol{\theta}, \boldsymbol{x})}\left[\log \frac{\hat{r}_{\boldsymbol{w}}(\boldsymbol{x} \,|\, \boldsymbol{\theta})}{Z_{\boldsymbol{w}}(\boldsymbol{x})}\right] \tag{43}$$

$$= I(\boldsymbol{\theta}; \boldsymbol{x}) - \mathbb{E}_{p(\boldsymbol{\theta}, \boldsymbol{x})}\left[\log \hat{r}_{\boldsymbol{w}}(\boldsymbol{x} \,|\, \boldsymbol{\theta})\right] + \mathbb{E}_{p(\boldsymbol{x})}\left[\log Z_{\boldsymbol{w}}(\boldsymbol{x})\right], \tag{44}$$

where $I(\boldsymbol{\theta}; \boldsymbol{x})$ is the *mutual information* w.r.t. $p(\boldsymbol{\theta}, \boldsymbol{x})$. Since we aim at minimizing the expected Kullback-Leibler divergence we implicitly aim to maximize our mutual information approximation:

$$I_{\boldsymbol{w}}^{(0)}(\boldsymbol{\theta}; \boldsymbol{x}) := \mathbb{E}_{p(\boldsymbol{\theta}, \boldsymbol{x})}\left[\log \hat{r}_{\boldsymbol{w}}(\boldsymbol{x} \,|\, \boldsymbol{\theta})\right] - \mathbb{E}_{p(\boldsymbol{x})}\left[\log Z_{\boldsymbol{w}}(\boldsymbol{x})\right] \tag{45}$$

$$= \mathbb{E}_{p(\boldsymbol{\theta}, \boldsymbol{x})}\left[\log \hat{r}_{\boldsymbol{w}}(\boldsymbol{x} \,|\, \boldsymbol{\theta})\right] - \mathbb{E}_{p(\boldsymbol{x})}\left[\log \mathbb{E}_{p(\boldsymbol{\theta})}[\hat{r}_{\boldsymbol{w}}(\boldsymbol{x} \,|\, \boldsymbol{\theta})]\right] \tag{46}$$

$$= \mathbb{E}_{p(\boldsymbol{\theta}, \boldsymbol{x})}\left[h_{\boldsymbol{w}}(\boldsymbol{\theta}, \boldsymbol{x})\right] - \mathbb{E}_{p(\boldsymbol{x})}\left[\log \mathbb{E}_{p(\boldsymbol{\theta})}[\exp(h_{\boldsymbol{w}}(\boldsymbol{\theta}, \boldsymbol{x}))]\right], \tag{47}$$

which can be estimated via Monte-Carlo by sampling i.i.d. $\boldsymbol{\theta}_n, \boldsymbol{\theta}_{n,m} \sim p(\boldsymbol{\theta})$, $\boldsymbol{x}_n \sim p(\boldsymbol{x} \,|\, \boldsymbol{\theta}_n)$, $n = 1, \ldots, N$, $m = 1, \ldots, M$ and then compute:

$$\hat{I}_{\boldsymbol{w}}^{(0)}(\boldsymbol{\theta}; \boldsymbol{x}) := \frac{1}{N}\sum_{n=1}^{N} \log \hat{r}_{\boldsymbol{w}}(\boldsymbol{x}_n \,|\, \boldsymbol{\theta}_n) - \frac{1}{N}\sum_{n=1}^{N} \log\left(\frac{1}{M}\sum_{m=1}^{M} \hat{r}_{\boldsymbol{w}}(\boldsymbol{x}_n \,|\, \boldsymbol{\theta}_{n,m})\right) \tag{48}$$

$$= \frac{1}{N}\sum_{n=1}^{N} h_{\boldsymbol{w}}(\boldsymbol{\theta}_n, \boldsymbol{x}_n) - \frac{1}{N}\sum_{n=1}^{N} \log\left(\frac{1}{M}\sum_{m=1}^{M} \exp(h_{\boldsymbol{w}}(\boldsymbol{\theta}_{n,m}, \boldsymbol{x}_n))\right). \tag{49}$$

Since in all mentioned methods $\hat{r}_{\boldsymbol{w}}(\boldsymbol{x} \,|\, \boldsymbol{\theta})$ is meant to approximate the ratio $\frac{p(\boldsymbol{\theta} \,|\, \boldsymbol{x})}{p(\boldsymbol{\theta})}$, and estimating the normalizing constant is expensive, a naive alternative to approximate the expected Kullback-Leibler divergence is by plugging the unnormalized distribution $\hat{q}_{\boldsymbol{w}}(\boldsymbol{\theta} \,|\, \boldsymbol{x}) := \hat{r}_{\boldsymbol{w}}(\boldsymbol{x} \,|\, \boldsymbol{\theta}) p(\boldsymbol{\theta})$ into the above formula and using the estimate:

$$\hat{I}_{\boldsymbol{w}}(\boldsymbol{\theta}; \boldsymbol{x}) := \frac{1}{N}\sum_{n=1}^{N} \log \hat{r}_{\boldsymbol{w}}(\boldsymbol{x}_n \,|\, \boldsymbol{\theta}_n) = \frac{1}{N}\sum_{n=1}^{N} h_{\boldsymbol{w}}(\boldsymbol{\theta}_n, \boldsymbol{x}_n). \tag{50}$$

While this is justified for the training objectives for NRE-A and NRE-C, which encourage a trivial normalizing constant $Z_{\boldsymbol{w}}(\boldsymbol{x}) \approx 1$ at optimum, the same is not true for NRE-B, which leads to an additional non-vanishing, possibly arbitrarily big, bias term:

$$\hat{I}_{\boldsymbol{w}}(\boldsymbol{\theta};\boldsymbol{x}) \approx I_{\boldsymbol{w}}^{(0)}(\boldsymbol{\theta};\boldsymbol{x}) + C_{\boldsymbol{w}} \qquad \text{(NRE-B)} \qquad (51)$$

Another way to address the normalizing constant is the use of the Kullback-Leibler divergence that also works for unnormalized distributions $p(\boldsymbol{z})$, $q(\boldsymbol{z})$:

$$\text{KLD}(p(\boldsymbol{z}) \,\|\, q(\boldsymbol{z})) := \int \left( \log \left( \frac{p(\boldsymbol{z})}{q(\boldsymbol{z})} \right) + \frac{q(\boldsymbol{z})}{p(\boldsymbol{z})} - 1 \right) p(\boldsymbol{z}) \, d\boldsymbol{z}, \qquad (52)$$

which is always $\geq 0$ with equality if $p(\boldsymbol{z}) = q(\boldsymbol{z})$ for $p(\boldsymbol{z})$-almost-all $\boldsymbol{z}$.

This gives for the expected Kullback-Leibler divergence between the posterior $p(\boldsymbol{\theta} \,|\, \boldsymbol{x})$ and the unnormalized approximate posterior $\hat{q}_{\boldsymbol{w}}(\boldsymbol{\theta} \,|\, \boldsymbol{x}) := \hat{r}_{\boldsymbol{w}}(\boldsymbol{x} \,|\, \boldsymbol{\theta})p(\boldsymbol{\theta})$:

$$\mathbb{E}_{p(\boldsymbol{x})} \left[ \text{KLD}(p(\boldsymbol{\theta} \,|\, \boldsymbol{x}) \,\|\, \hat{q}_{\boldsymbol{w}}(\boldsymbol{\theta} \,|\, \boldsymbol{x})) \right] \qquad (53)$$

$$= \mathbb{E}_{p(\boldsymbol{\theta},\boldsymbol{x})} \left[ \log \frac{p(\boldsymbol{\theta} \,|\, \boldsymbol{x})}{\hat{q}_{\boldsymbol{w}}(\boldsymbol{\theta} \,|\, \boldsymbol{x})} + \frac{\hat{q}_{\boldsymbol{w}}(\boldsymbol{\theta} \,|\, \boldsymbol{x})}{p(\boldsymbol{\theta} \,|\, \boldsymbol{x})} - 1 \right] \qquad (54)$$

$$= \mathbb{E}_{p(\boldsymbol{\theta},\boldsymbol{x})} \left[ \log \frac{p(\boldsymbol{\theta},\boldsymbol{x})}{p(\boldsymbol{\theta})p(\boldsymbol{x})} \frac{p(\boldsymbol{\theta})}{\hat{q}_{\boldsymbol{w}}(\boldsymbol{\theta} \,|\, \boldsymbol{x})} + \frac{\hat{q}_{\boldsymbol{w}}(\boldsymbol{\theta} \,|\, \boldsymbol{x})p(\boldsymbol{x})}{p(\boldsymbol{\theta},\boldsymbol{x})} - 1 \right] \qquad (55)$$

$$= I(\boldsymbol{\theta};\boldsymbol{x}) - \mathbb{E}_{p(\boldsymbol{\theta},\boldsymbol{x})} \left[ \log \hat{r}_{\boldsymbol{w}}(\boldsymbol{x} \,|\, \boldsymbol{\theta}) \right] + \int \left( \int \hat{r}_{\boldsymbol{w}}(\boldsymbol{x} \,|\, \boldsymbol{\theta}) \, p(\boldsymbol{\theta}) \, d\boldsymbol{\theta} \right) p(\boldsymbol{x}) \, d\boldsymbol{x} - 1 \qquad (56)$$

$$= I(\boldsymbol{\theta};\boldsymbol{x}) - \mathbb{E}_{p(\boldsymbol{\theta},\boldsymbol{x})} \left[ \log \hat{r}_{\boldsymbol{w}}(\boldsymbol{x} \,|\, \boldsymbol{\theta}) \right] + \mathbb{E}_{p(\boldsymbol{x})} \left[ Z_{\boldsymbol{w}}(\boldsymbol{x}) - 1 \right]. \qquad (57)$$

We see that the normalizing constant $Z_{\boldsymbol{w}}(\boldsymbol{x})$ re-appears, but with a different term. Similar to before the above can be used for another mutual information approximation given by:

$$I_{\boldsymbol{w}}^{(1)}(\boldsymbol{\theta};\boldsymbol{x}) := \mathbb{E}_{p(\boldsymbol{\theta},\boldsymbol{x})} \left[ \log \hat{r}_{\boldsymbol{w}}(\boldsymbol{x} \,|\, \boldsymbol{\theta}) \right] - \mathbb{E}_{p(\boldsymbol{x})} \left[ Z_{\boldsymbol{w}}(\boldsymbol{x}) - 1 \right] \qquad (58)$$

$$= \mathbb{E}_{p(\boldsymbol{\theta},\boldsymbol{x})} \left[ \log \hat{r}_{\boldsymbol{w}}(\boldsymbol{x} \,|\, \boldsymbol{\theta}) \right] - \mathbb{E}_{p(\boldsymbol{x})p(\boldsymbol{\theta})} \left[ \hat{r}_{\boldsymbol{w}}(\boldsymbol{x} \,|\, \boldsymbol{\theta}) - 1 \right] \qquad (59)$$

$$= \mathbb{E}_{p(\boldsymbol{\theta},\boldsymbol{x})} \left[ h_{\boldsymbol{w}}(\boldsymbol{\theta},\boldsymbol{x}) \right] - \mathbb{E}_{p(\boldsymbol{x})p(\boldsymbol{\theta})} \left[ \exp(h_{\boldsymbol{w}}(\boldsymbol{\theta},\boldsymbol{x})) - 1 \right], \qquad (60)$$

which can be estimated via Monte-Carlo, again, by sampling i.i.d. $\boldsymbol{\theta}_n, \boldsymbol{\theta}_{n,m} \sim p(\boldsymbol{\theta})$, $\boldsymbol{x}_n \sim p(\boldsymbol{x} \,|\, \boldsymbol{\theta}_n)$, $n = 1, \ldots, N$, $m = 1, \ldots, M$ and then computing:

$$\hat{I}_{\boldsymbol{w}}^{(1)}(\boldsymbol{\theta};\boldsymbol{x}) := \frac{1}{N} \sum_{n=1}^{N} \log \hat{r}_{\boldsymbol{w}}(\boldsymbol{x}_n \,|\, \boldsymbol{\theta}_n) - \frac{1}{N} \frac{1}{M} \sum_{n=1}^{N} \sum_{m=1}^{M} \left( \hat{r}_{\boldsymbol{w}}(\boldsymbol{x}_n \,|\, \boldsymbol{\theta}_{n,m}) - 1 \right) \qquad (61)$$

$$= \frac{1}{N} \sum_{n=1}^{N} h_{\boldsymbol{w}}(\boldsymbol{\theta}_n, \boldsymbol{x}_n) - \frac{1}{N} \frac{1}{M} \sum_{n=1}^{N} \sum_{m=1}^{M} \left( \exp(h_{\boldsymbol{w}}(\boldsymbol{\theta}_{n,m}, \boldsymbol{x}_n)) - 1 \right). \qquad (62)$$

Note that since $\log(r) \leq r - 1$ we always have the inequalities:

$$I(\boldsymbol{\theta};\boldsymbol{x}) \geq I_{\boldsymbol{w}}^{(0)}(\boldsymbol{\theta};\boldsymbol{x}) \geq I_{\boldsymbol{w}}^{(1)}(\boldsymbol{\theta};\boldsymbol{x}), \qquad\qquad \hat{I}_{\boldsymbol{w}}^{(0)}(\boldsymbol{\theta};\boldsymbol{x}) \geq \hat{I}_{\boldsymbol{w}}^{(1)}(\boldsymbol{\theta};\boldsymbol{x}), \qquad (63)$$

showing that $I_{\boldsymbol{w}}^{(0)}(\boldsymbol{\theta};\boldsymbol{x})$ leads to a tighter approximation to the mutual information $I(\boldsymbol{\theta};\boldsymbol{x})$ than $I_{\boldsymbol{w}}^{(1)}(\boldsymbol{\theta};\boldsymbol{x})$.

The procedures above require estimating the normalizing constant or the partition function. Generally this can be quite expensive and may require techniques like Nested Sampling [65]; however, it is tractable with Monte Carlo on problems with parameters within low dimensional compact regions. That being said, the ratio can also introduce large variance in the integral estimates. This occurs when the posterior is much narrower than the prior, i.e., when data $\boldsymbol{x}$ carries a lot of information about $\boldsymbol{\theta}$.

**Bounds on the mutual information**  There is a connection between the training objective of NRE-B and a multi-sample lower bound on the mutual information [58], as noted in Durkan et al. [16]. Contrastive learning has been explored for estimating the mutual information by Van den Oord et al. [71] also discussed by Belghazi et al. [4]. The bounds we define above are also discussed in detail by Poole et al. [58], although we find computing $\hat{I}_{w}^{(0)}(\boldsymbol{\theta}; \boldsymbol{x})$ and $\hat{I}_{w}^{(1)}(\boldsymbol{\theta}; \boldsymbol{x})$ to be tractable within SBI–although potentially expensive and high-variance with an extremely narrow posterior.

We attempted to train a ratio estimator by minimizing $\hat{I}_{w}^{(1)}(\boldsymbol{\theta}; \boldsymbol{x})$ directly on fixed data. Our preliminary experiments found that the C2ST was near unity on the SLCP task and other works claim that this mutual information bound has extremely high variance. We therefore ended our investigation.

**Numerical estimates of bounds on mutual information**  In remains unclear how to evaluate the performance of SBI algorithms across model types without access to the ground truth. Computing $-\hat{I}_{w}^{(0)}(\boldsymbol{\theta}; \boldsymbol{x})$ as a validation loss is applicable to NRE-B and NRE-C for all $\gamma$ and $K$. Therefore, we investigate estimating this bound on the mutual information for model comparison and as a surrogate for computing the C2ST across several pieces of simulated data. (It is also noteworthy that this bound is applicable to Neural Posterior Estimation, where the likelihood-to-evidence ratio would be approximated by $p_{w}(\boldsymbol{\theta} \mid \boldsymbol{x})/p(\boldsymbol{\theta})$ and $p_{w}(\boldsymbol{\theta} \mid \boldsymbol{x})$ represents an approximate posterior density. However, this case is not investigated further in this work.)

In the effort to find a model comparison metric that applies when the user does not have access to the ground truth, we ran another set of experiments where we trained ratio estimators using NRE-B and NRE-C over various $\gamma$ and $K$, then we validated them on held-out data with $-\hat{I}_{w}^{(0)}(\boldsymbol{\theta}; \boldsymbol{x})$ as a validation loss. All networks corresponded with the *Large NN* architecture. Just like in the main experiments, we computed the C2ST over the ten different observations from the SBI benchmark. The results of the training can be seen in Figure 4 and in full in Figure 12. Visually, $-\hat{I}_{w}^{(0)}(\boldsymbol{\theta}; \boldsymbol{x})$ was more comparable across models than plotting their classification validation loss, see Figure 10, for both NRE-B and NRE-C. The classification validation loss has a bias depending on $\gamma$ and $K$ which $-\hat{I}_{w}^{(0)}(\boldsymbol{\theta}; \boldsymbol{x})$ does not exhibit. The metric therefore allows us to compare models such that model producing the most negative mutual information bound also most-accurately estimates the posterior, on average.

A correlation plot showing the relationship between $\hat{I}_{w}^{(0)}(\boldsymbol{\theta}; \boldsymbol{x})$ and the C2ST for various $\gamma$ and $K$ on the SLCP task can be found in Figure 13. We find that the two measurements are well correlated, implying that $-\hat{I}_{w}^{(0)}(\boldsymbol{\theta}; \boldsymbol{x})$, which does not require access to the ground truth posterior, may be able to replace computing the C2ST across several pieces of data, which does require access to the ground truth posterior, but further investigation is necessary.

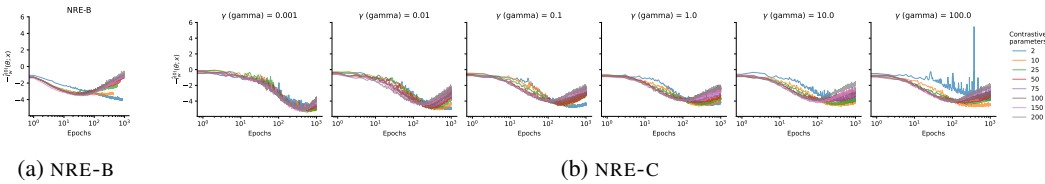

(a) NRE-B                                          (b) NRE-C

Figure 12: Our proposed metric, a negative bound on the mutual information $-\hat{I}_{w}^{(0)}(\boldsymbol{\theta}; \boldsymbol{x})$, for the SLCP task estimated over the validation set versus training epochs using (a) NRE-B and (b) NRE-C with various values of $\gamma$ and $K$, a Large NN architecture, and fixed training data. The bound permits visualization of the convergence rates and pairwise comparison across models.

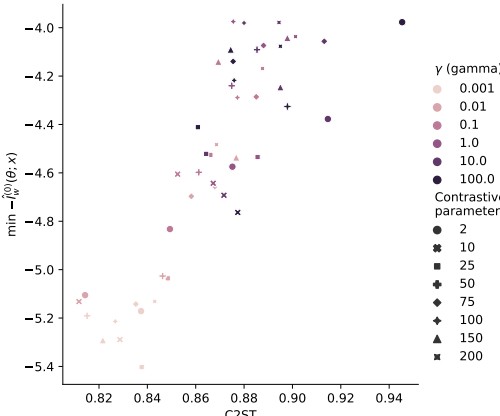

Figure 13: A scatter plot of the minimum $-\hat{I}_{\boldsymbol{w}}^{(0)}(\boldsymbol{\theta}; \boldsymbol{x})$ versus the C2ST on the SLCP task with every point corresponding to a different set of values for $\gamma$ and $K$. C2ST scales from 0.5 to 1.0 with 0.5 implying that the classifier could not distinguish the approximate posterior from the ground truth. Just like in the fixed data regime in the main experiment, see Figure 6, we found that on this task a lower $\gamma$ improved the C2ST. We also find that the mutual information is correlated with the average C2ST across 10 pieces of data, but the mutual information has the practical advantage that we can bound it without knowing the ground truth posterior. The C2ST requires being able to sample from the ground truth posterior. This data represents the same set of experiments as in Figure 4, Figure 12, and Figure 14.

**Numerical estimates of the partition function**   In addition to bounds on the mutual information, we computed a Monte Carlo estimate of the partition function, $Z_{\boldsymbol{w}}(\boldsymbol{x})$, based on data from the validation set at every epoch during training. The value of the estimated partition function as a function of epoch is shown in Figure 14 for this set of runs on the SLCP task. The estimated $Z_{\boldsymbol{w}}(\boldsymbol{x})$, based on the ratio from NRE-B, is completely unconstrained and varies significantly with epoch. NRE-C does encourage the partition function to remain "near" unity, although both $\gamma$ and $K$ affect the strength of the encouragement. We find that large numbers of contrastive parameters cause $Z_{\boldsymbol{w}}(\boldsymbol{x})$ to deviate significantly from unity; although, often by tens of orders of magnitude less than NRE-B.

This result is connected to the reliability of the importance sampling diagnostic, see Section 2.2. If the partition function is not near unity, then the estimated likelihood-to-evidence ratio does not cancel with the evidence and the diagnostic will behave like NRE-B, i.e., it becomes possible to produce accurate, albeit unnormalized, posteriors while failing the diagnostic. This is one reason to limit the number of contrastive parameters.

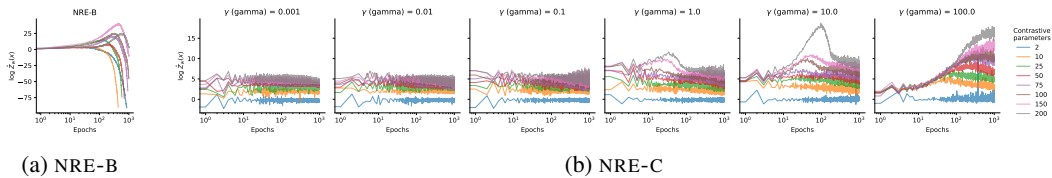

(a) NRE-B                                                    (b) NRE-C

Figure 14: Monte Carlo estimate of the partition function $\hat{Z}_{\boldsymbol{w}}(\boldsymbol{x})$ for (a) NRE-B and (b) NRE-C with various values for $\gamma$ and $K$ on the SLCP inference problem. Note the scale of the ordinate axes. $\hat{Z}_{\boldsymbol{w}}(\boldsymbol{x})$ is completely unconstrained with NRE-B; however, it is constrained with NRE-C. The strength of encouragement the partition function towards unity depends on the hyperparameters for NRE-C. $\hat{Z}_{\boldsymbol{w}}(\boldsymbol{x})$ can still take on extreme values, especially for large number of contrastive examples and large $\gamma$. This represents the same set of experiments as in Figure 4, Figure 4 and Figure 13.