# OpenReview forum: "Contrastive Neural Ratio Estimation"
_NeurIPS.cc/2022/Conference — NeurIPS 2022 Accept_

### Official Review · Reviewer_fMcW · 2022-06-17

**Rating:** 6
**Confidence:** 2
**Soundness:** 4 excellent
**Presentation:** 2 fair
**Contribution:** 3 good

**Summary:**

Authors present a generalising framework of (what they call) NRE-A and NRE-B. They argue that is has clear advantages over especially NRE-B. In the experiment section they show some good values of hyperparameters K and gamma, found on a benchmark dataset.

**Questions:**

I really enjoyed your introduction and motivation; it helped me a lot. However, I could understand that equation 2 is not influenced by k? Can you elaborate here?

In general, I think the motivation to have multiple k could be more elaborate. You mention it improves simulation-efficiency; is there more to it?

Line 89: "In contrast with NRE-A, one of these parameters θ_k is always drawn jointly with x.". As far as I could tell one of the two classes in NRE-A is exactly drawn jointly with x? So why is it in contrast?

In Equation 14: is Z_w computable? Or should it be approximated, and how good can such an approximation be? And, in data-limit (only) how quick is converge to 1? Vague answer is accepted here.




**Limitations:**

The authors include a section on societal limitations. I agree completely with the authors here.

**Strengths And Weaknesses:**

I think the major strength is the generalising framework the authors present. This type of work usually helps fields move forward.
The weakness, in my eyes, is the introduction of hyperparameters, especially gamma. I am unsure how easy it is to find good values of this if data is limited.

---

> ### Author Response · Authors · 2022-07-30
> **iniital reply to fMcW - 02**
>
> ### Questions
>
> -   You are correct that equation (2) is not influenced by K and we’re happy to discuss it further! Equation (2) represents the optimal ratio estimated by optimally minimizing the NRE-B loss function with an infinitely flexible neural network. It is not a function of K because no matter the value of K >=1, the optimal minimizer of the NRE-B loss converges to the same ratio. The paper introducing NRE-B does an empirical study of how changing K affects the convergence rate with local optimizers, finite networks, and limited training data. (Just like we do in our paper!) Please let us know if this adequately clarifies the question.
>
> -   The motivation for the investigation was primarily to investigate the properties of a generalization of these two methods. Motivation for increasing the number of contrastive examples comes partially from the introduction of the idea in the paper by Durkan, et. al., but also is a natural extension of a binary classifier and goes along with advances in contrastive representation learning where the number of classes can affect performance. We also emphasize that the weighting of the classes with gamma is equally explored in our paper, both of these searches come naturally from the generalizing framework that we found for NRE-A and NRE-B. We report the simulation efficiency along with other empirical results in different regimes.
>
>     We claim this simulation-efficiency motivation is inherited from Durkan, but we want to clarify what we mean there. Durkan’s paper is primarily designed to point out a connection between posterior estimation and ratio estimation and introduce a “more simulator efficient” sequential algorithm. We changed our language to read `As a part of an effort to unify different \SBI methods and to improve simulation-efficiency, \citet{Durkan2020} reformulated the classification task to identify which of $K$ possible $\btheta_k$ was responsible for simulating $\bx$.`
>
> -   Excellent find! Indeed, that is a typo. We changed it to read `\citet{Durkan2020} train a classifier that selects from among $K$ parameters $(\btheta_1, \ldots, \btheta_K)$ which could have generated $\bx$, in contrast with \NREA's binary possibilities. One of these parameters $\btheta_k$ is \emph{always} drawn jointly with $\bx$.`
>
> -   Great question! Although (14) is generally intractable to compute exactly, it is possible to estimate (14) using Monte Carlo. We do exactly that for NRE-B and NRE-C in the new version of Appendix D, Figure 13. Estimating the partition function (this quantity) is an entire field of research within statistical physics and energy based models. Nested sampling provides one method, but there are many different possibilities. Since we can easily sample from the joint distribution, we computed it using simple Monte Carlo (this is generally available outside of SBI).
>
>
> ### Conclusion
>
> Sharp eye finding those typos and asking informative questions to improve the paper! We appreciate your time and expertise in your reply. If you found our changes adequate we hope you’ll consider improving your score to reflect the improvement in the paper itself.

---

> > ### Comment · Reviewer_fMcW · 2022-08-08
> > **Thank you**
> >
> > I thank the authors for their clarifications of my issues and misunderstandings. I remain positive about this paper, and I keep my score. I suggest acceptance of this paper.

---

> ### Author Response · Authors · 2022-07-30
> **initial reply to fMcW - 01**
>
> We appreciate the time taken to review our paper and the insights that you provided with your discussion. We’re glad about the positive assessment of the paper and wish to clarify any issues in our response!
>
> ### Summary
>
> -   We find your summary to be accurate, thank you for reading and responding to the paper! We do see improvement over NRE-A in some contexts as well, in particular the infinite joint draws in the C2STs hyperparameter search.
>
>
> ### Strengths And Weaknesses
>
> -   Indeed we agreed that the generalization of the two methods is our fundamental contribution and we are glad that you also find it to be a step forward. You are correct that we introduce the two hyperparameters, but we see them as “interpolating” between NRE-A and NRE-B. I.e. before it was only possible to select two options from these hyperparameters, now any selection is allowed.
>
>     We hope that our recommendations about how to set the hyper parameters will be useful to practitioners as they improve the performance of the algorithm across the SBI benchmark. We have expanded this recommendation in the revised version of our paper and also been more specific when certain hyperparameters might be valuable--primarily discussing the effects on the diagnostic and normalizing constant. Higher K increases the normalizing constant and gamma reduces convergence rate but can sometimes lead to higher accuracy models, generally gamma=1.0 is a safe choice.
>
>     In SBI data is only limited by the simulation budget of the user. Therefore, if our suggested hyperparameters are not satisfactory for a user; a hyperparameter search like we conducted should be possible in the SBI setting.

---

### Official Review · Reviewer_1x3V · 2022-06-29

**Rating:** 8
**Confidence:** 3
**Soundness:** 3 good
**Presentation:** 3 good
**Contribution:** 3 good

**Summary:**

Density-ratio estimation can be performed via classification tasks (binary, multisample) and learn a potentially unnormalized model for the posterior distribution.

This submission highlights a simple yet important flaw: when the posterior is parameterized as a softmax (this is the case for NRE-B a.k.a. InfoNCE), the scaling indeterminacy translates to a bias in the density-ratio estimate which trickles down to the rest of the analysis. The submission proposes a new loss that removes this bias. It carefully examines how that impacts posterior estimation in practice, through a series of controlled experiments.


**Questions:**

It is known from [1] that the NRE-B loss recovers the density ratio + a bias term. However, in [1] the bias is learnt as part of the estimation procedure. This way, the *un-normalized* density ratio (and by extension, its numerator the conditional density model, in your case the posterior) can be recovered without bias.

Could you clarify why learning the bias does not solve the problem? Is it because you would only recover the *unnormalized* posterior whereas you need the *normalized* posterior in the diagnostics?

**Ethics Review Area:**

["I don’t know"]

**Limitations:**

I see no particular negative societal impact.

**Strengths And Weaknesses:**

**Strengths**

This work is interesting, as it highlights a simple yet important flaw in a popular loss (InfoNCE) and shows how a simple fix can lead enable better posterior learning.

The connections with NRE-A (essentially NCE, depending on how the discriminator is parameterized) and NRE-B (InfoNCE) are appreciated. Given that the authors' contribution, NRE-C, can be seen as a generalization of the previous losses, it is interesting to see in Figure 1 what the best setup is: while it has been noted in previous work that (i) using many negative samples are advantageous, the insight that (ii) imbalanced classification is favorable when the tuples of entirely noise samples are over-represented, seems novel.

The diagnostics section is appreciated as it is an important and rather open question when estimating conditional density models, such as the posterior distribution.

**Weaknesses**

As I understand it, the simulations evaluate the quality of estimation by averaging across datasets (Figure 4). Is it common to report the variance *across* datasets? Could you report the variance *per* dataset across many (data-generation) *seeds*?

The effect of the hyperparameters on the convergence speed (Figure 5) is interesting, however it is not clear to me from the plot "that increasing K leads to earlier convergence at fixed gamma". Could the authors explain further?

---

> ### Author Response · Authors · 2022-07-30
> **initial reply to 1x37 - 02**
>
> ### Questions
>
> -   We are not quite sure what is meant by [1]. If you mean “massive optimal data compression…” by J. Alsing, et. al. We’re not sure where in the paper they learn the partition function (c_w(x)). Perhaps the link was forgotten?
>
>     To answer your question, you are correct that learning the unnormalized posterior is not suitable for the importance sampling diagnostic. If it were possible to learn the c_w(x), then it seems likely that this could de-bias the estimate and the result `log \frac{p(x \mid \theta)}{p(x)} + c_w(x) - \hat{c}_w(x)` (with the estimate the hat) could be closer to a normalized posterior. Yet unless `c_w(x) - \hat{c}_w(x)` is nearly constant over all x, then we will still have the issues we raise in Appendix B.2
>
>     We show in appendix B.2 that the unknown bias drops out only when C(x_n) \approx C(x_i) for all i and n, which occurs only when C is nearly constant. Since NRE-B does not restrict it, that means C is probably not constant in x and is unlikely to drop out by normalizing (although learning it may bring this debiased estimate closer to constant). This means the importance sampling diagnostic will still fail after normalizing. For this reason, we added the sentence in the “importance sampling diagnostic” paragraph `... with normalized importance weights is ill-posed, i.e., the problem is \emph{not} solved by estimating the partition function.` We also added a similar sentence to appendix B.2.
>
>     Furthermore, NRE-B does not outperform NRE-C in our investigations. Therefore, we still would recommend using NRE-C. We show in appendix D in an updated version of the paper that the mean E[c_w(x)] in NRE-B can get quite large, thus we wonder whether learning it is a better approach than just encouraging c_w(x) \approx 1 from the start, as we do in NRE-C
>
>
>
>
> ### Conclusion
>
> We thank you again for your important clarifying questions. We are happy to discuss any further questions or follow up if you find that our response could use further explanation.

---

> > ### Comment · Reviewer_1x3V · 2022-08-08
> > **Response to authors**
> >
> > I appreciate the authors' responsiveness! Apologies for the delay.
> >
> > - Updated results table
> >
> > Thank you for providing the intermediate results: I understand the experiments are still running over 10 seeds and I trust the table with the variability across seeds *per* task will be added to Appendix C in the final version, as the authors said. Though it might not change the overall conclusion, I think this allows to check that there is not some "compensation effect" across tasks and that the "source of variability" is indeed sampling and training (different seeds, posterior models, stochastic optimization).
> >
> > - Clarification on sequential vs. amortized
> >
> > As far as my understanding goes, the big picture is you're comparing different methods for estimating a posterior for SBI. You do that by comparing statistical properties (averaging across seeds in the table), optimization properties (convergence rates, Fig 5), and you mention that computational properties (amortized vs. sequential) are important as well and favor NRE-C over others methods in SBI. I am unclear however on which methods are amortized (all the NREs?) and which are sequential. This is discussed at a conceptual level in the end of the intro + the appendix, but it is not clear to me for which models and at which steps this specifically intervenes in the comparisons you're carrying out. I think it is worth clarifying, especially if your model (and the other NREs) benefit from being amortized.
> >
> > - Writing and relevant literature
> >
> > I appreciate the authors are expanding their referencing of relevant literature: links between SBI (NRE-A, NRE-B), Estimation (NCE [1], RankingNCE [2]), and Representation Learning (NCE and Negative sampling [3, 4], InfoNCE [5]) are longstanding. These links have been noted for SBI in [6]. I think this should be added to the introduction (instead of the Appendix) as it is relevant literature: it positions NRE-C in a wider context.
> >
> > - Clarity of Fig 5
> >
> > Too many contrastive parameters (all the different values of K) as well as different panels (different gammas) makes it hard to read. I would keep only two panels: left NRE-B, right NRE-C. The NRE-C single panel (bigger, given more space) would have: gamma = 1e-3 and 1 and 100 (1e-3 and 1 are the important values given Fig 1) denoted with a hue (e.g. triangle, circle, and cross markers, or different linestyles), and K=2 and 150 with two colors (e.g. blue and green).
> >
> > The rest could be included in the Appendix.
> >
> > - Comparing NRE-B and NRE-C
> >
> > In my view, this is very important to discuss further as it is the premise of this paper: c_w(x) is problematic and NRE-C avoids that altogether by changing the loss.
> > For NRE-B, as you said, the Importance Sampling diagnostic fails because the bias C(x) is not constant. However as you said, *you could learn C(x)* which should de-bias the estimation of w: this is exactly what is done in [2] for RankingNCE (your NRE-B), at the end of Assumption 2.2! However, this works under certain identifiability conditions, see Assumption 4.1 in [2], which seems to connect with what you are saying "c_w(x) - \hat{c}_w(x) is nearly constant over all x" if I am not mistaken.
> > If you believe this assumption is unrealistic and that this fragilizes NRE-B as the estimate of w depends on how well C_w(x) is estimated, that is worth saying. Or if you think this should be investigated in future work, that is worth saying as well.
> >
> > NRE-B or InfoNCE is the powerhorse of many practical breakthroughs in contrastive learning and has been applied to representation learning, SBI, and estimation. However it relies on being able to estimate c_w(x) adequately and this discussion has largely been ignored. Because NRE-B is so widely used, this makes the discussion on c_w(x) important to bring to the table and it directly influences the estimation of w and its diagnostic. This discussion on learning c_w(z), how its estimation impacts that of w, how realistic it is to expect that it is well estimated, and referencing the prior work [2] which exactly formalizes that, is a perfect discussion point. And it provides a way to frame your method NRE-C which basically says: instead of dealing with the c_w(x) term in NRE-B, we can change the loss to remove that obstacle.
> >
> >
> > [1] Gutmann and Hyvarinen 2012 JMLR
> >
> > [2] Ma and Collins 2018 (Noise Contrastive Estimation and Negative Sampling...)
> >
> > [3] Mnih and Teh, 2011
> >
> > [4] Mikolov et al. 2013 (Distributed Representations of Words...)
> >
> > [5] Van den Oord et a. 2018 (Representation Learning with Contrastive Predictive Coding)
> >
> > [6] On Contrastive Learning for Likelihood-free Inference. Conor Durkan, Iain Murray, George Papamakarios.

---

> > > ### Comment · Reviewer_1x3V · 2022-08-08
> > > **Response to authors (part 2)**
> > >
> > > **Conclusion**: I trust the reviewers will address the points above in the final version of the paper. Apart from the updated table which they have already started computing, the remaining points are text and figures (easily actionable) however I believe they are necessary to add substantial clarity, more complete references, and important discussion points to the final version. On that basis, I have raised my score as I think this is an interesting contribution:
> > > - for different fields (SBI, Contrastive Learning, Density-Ratio Estimation)
> > > - which raises interesting future questions, e.g. the impact of K >> 1 on statistical properties (robustness of the estimation to range of gamma values) and optimization properties (quicker convergence)

---

> ### Author Response · Authors · 2022-07-30
> **initial reply to 1x37 - 01**
>
> Thank you for reading our paper and providing a lot of insight into its relationship with noise contrastive estimation. We also appreciate that your questions and requests were highly actionable. We found our paper has improved by incorporating your suggestions, we hope you think so too!
>
> ### Summary
>
> -   We agree with the summary and want to note that the reviewer even pointed out an important connection to NCE and InfoNCE explicitly, which is quite nice. We are expanding our references to the contrastive learning literature and we will make sure this is made explicit in the text. We added a sentence about it in appendix A. `Specific connections are in the loss functions with \NREA closely corresponding to noise-contrastive estimation (NCE) \cite{gutmann2010noise, gutmann2012noise} and \NREB with InfoNCE \cite{van2018representation}.`
>
>
> ### Strengths And Weaknesses
>
> Strengths
>
> -   We appreciate the compliments on the strengths and the connections with contrastive learning. We are happy to see that the strengths of our method align with what we see as our main contributions.
>
>
> Weaknesses
>
> -   You correctly pointed out that we average across data for every individual task and also across tasks (task averaging is not addressed in our response here). In this case, the data average is over 10 different simulated posteriors, namely the mean C2ST reported is computed 1/N \sum_{i=1}^N C2ST[p_w(\theta \mid x_i), p(\theta \mid x_i)] with N=10.
>
>     In the SBI benchmark it is computed 1/N \sum_{i=1}^N C2ST[p_{w_i}(\theta \mid x_i), p(\theta \mid x_i)] where w_i implies that the weights have been trained differently due to different seeds (and also different simulated training data).
>
>     That means the SBI benchmark trains 10 times the estimators that we do, generates 10 times the amount of training data, and averages over 10 posteriors for each observation. One huge advantage of NRE-C’s amortized estimation is the ability to train just one estimator and average over 10 different posteriors. (Many estimators in the SBI benchmark are sequential and do NOT have this property.) We find this way of averaging to be in line with the mutual information bound that we apply in figure 5 in the revised version of the paper, i.e. it tells us about the performance of the specific estimator. That’s why we did it this way.
>
>     That being said, we understand that also averaging over seed is also an interesting result to present. We just started running the benchmark again, that way we can report an average with a similar evaluation technique to the SBI benchmark: over seed AND data. When it is finished, we will update our tables to represent this averaging over both seed and data (just like in the sbi benchmark.) When it’s done, we will add clarifying information about this to Appendix C.2. We do not expect significant changes in the reported results due to the averaging across posteriors (data) and tasks (in the main c2st result table).
>
> -   We believe that indeed the plot is a bit hard to read. We think the lack of clarity is due to the offset between different values of K making it difficult to compare the minima of different colors (aka K) in the same plot (same gamma). In the new version of our paper we use a bound on the mutual information to validate models and select between them where the offsetting does not appear.
>
>     When gamma=0.001 the effect is very small, arguably non-existent (all models converge around 5*10^2 epochs). However, in all other plots the grey line (200 contrastive parameters) achieves a minimum in fewer epochs (earlier convergence) than any other colored line (sometimes it looks more-or-less tied with pink). Hopefully this clarifies what we mean. If it is not clarified and you think we should soften / change our language, please let us know! For now, we changed the sentence in the figure so it reads`A grid search indicates that increasing $K$ leads to earlier convergence at fixed $\gamma$, when $\gamma > 0.001$.`

---

> ### Author Response · Authors · 2022-08-04
> **notification of new results**
>
> We included new results in the table displayed at the link:
> https://openreview.net/forum?id=kOIaB1hzaLe&noteId=-_v160043W
>
> As requested, we trained from 4 different seeds and present the average in the table. We will continue the experiments to get to 10, but some tasks are rather slow.

---

### Official Review · Reviewer_gFMN · 2022-07-12

**Rating:** 6
**Confidence:** 4
**Soundness:** 3 good
**Presentation:** 3 good
**Contribution:** 2 fair

**Summary:**

This paper introduces contrastive neural ratio estimation (NRE-C) to develop a better estimator for the likelihood-to-evidence ratio p(x | \theta)/p(x) for simulation-based inference (SBI). They first cast the parameter estimation problem as a classification problem, then augment the K-class classification problem (where the goal is to select the best parameter \theta_k among K options) in Durkan et al. with another (K+1)th class which contains samples that were generated independently from p(\theta)p(x). They demonstrate that NRE-C is consistent and outperforms existing baselines on simulated benchmarks.

**Questions:**

- When constructing this “hybrid” objective function (that blends NRE-A and NRE-B), what exactly are the limitations? For example, is it computationally expensive to generate samples x^{(b)} \sim p(x)p(\theta) to sZet up the classification problem? Given that the simulator takes a prior p(\theta) and a way to sample from the likelihood p(x|\theta), it seems like sampling from p(x) would be very hard). The paper also mentioned that sampling from the prior is slow. If we take into consideration this samping procedure for constructing the (K+1)th class, is NRE-C still more efficient than NRE-B?

**Limitations:**

Broader Impact statement is too broad, should be more specific about what “decisions about matters that affect living creatures” are, etc.

**Strengths And Weaknesses:**

Strengths:
- NRE-C is a clean and straight-forward generalization of Durkan et al. and the binary classification “trick” for estimating likelihood ratios.
- The authors also performed extensive experiments and empirical evaluations on the SBI benchmarks suite.

Weaknesses:
- It’d be helpful if the paper was a bit more self-contained. There is a lot of information in the supplementary that would benefit from being moved to the main text (e.g. related works, further details on the SBI benchmark). As another example, l. 35 (equivalence of the 2 ratios) should be better explained. Why do we want these two quantities to be equivalent? The text should explicitly mention that it is to try to make posterior inference easier in SBI.
- The presentation was also unclear. For example, the figures are pretty confusing – I’m not sure what we are supposed to be taking away from this image/caption. Figure 1 was unclear, and Figure 2 was also not clear with the different notations (e.g. b, b’).
- There is notation (e.g. f_w, x, theta, theta_0, p, etc.) that is not explained. It’d be clearer if this notation was collected in the following section (or introduced properly before its usage) to make it easier to follow. This causes conduction, such as \sigma \circ f_w(\theta, x) on l. 41.

---

> ### Author Response · Authors · 2022-07-30
> **initial reply to gFMN - 01**
>
> Thank you very much for your time and insight into our work. We hope the changes we made to the paper address your concerns. We see the changes as helping clarify the message.
>
> ### Summary
>
> - Thank you for the clear and accurate summary.
>
>
>
> ### Strengths And Weaknesses
>
> Weaknesses
>
> -   We acknowledge the point that the work is not completely self contained, especially regarding the details of the related work and simulation-based inference benchmark.
>
>
> -   We added more detail regarding the relevant simulators from the benchmark to the appendix C.2. Given space, we can improve the descriptions with more detail in the “experiment” section in paragraph 2. (Although there is not much space now. Upon acceptance with another content page, we try to fit this.)
>
> -   We agree that moving some of the information about related work, such as sequential / deep learning approaches, Into the main text (intro) would be helpful. (Upon acceptance we can use some of the additional content page for this.)
>
> -   To make the relationship between the ratios clearer we introduced a sentence to the intro `an appealing alternative for practitioners is estimating a \emph{ratio} between distributions. Specifically, the likelihood-to-evidence ratio $\frac{p(\btheta \mid \bx)}{p(\btheta)} = \frac{p(\bx \mid \btheta)}{p(\bx)} = \frac{p(\btheta, \bx)}{p(\btheta) p(\bx)}$. It relates the prior to the posterior as can be shown using Bayes' rule.`
>
> -   In order to make it clearer that we want to estimate the posterior efficiently, we added the following sentence to the first introductory paragraph `Our design goal is to produce a surrogate model $\phat(\btheta \mid \bx)$ approximating the posterior for any data $\bx$ while limiting excessive simulation.`
>
>
> -   We are glad that you brought up these concerns because we think that Figure 1 and Figure 2 are very important to make clear to the reader. Figure 1 sketches the performance of our algorithm over the hyperparameters we investigate in the paper. Figure 2 shows the relationship between our general NRE-C compared to the two previous methods NRE-A and NRE-B.
>
>
> -   We changed the first sentence in Figure 1 to `Conceptual, interpolated map from investigated hyperparameters of proposed algorithm \NREC to a measurement of posterior exactness using the Classifier Two-Sample Test.` Hopefully this helps to clarify its purpose. Please let us know if you want something else specific to be changed to help clarify the plot.
>
> -   It seems that the issue with Figure 2 is the lack of clarity in the notation in the loss functions. To solve that issue we added the sentence `Notation is defined in Section~\ref{sec:nrec}.` which refers to where the notation is introduced. We attempted to define the relevant symbols in the caption but it took up too much space. Please let us know if that addresses your concerns or if there is another way we can clarify the figure.
>
>
> -   You are absolutely correct that the construction with sigma \circ f_w is used before f_w or \circ are introduced. We have fixed this by explaining what those symbols represent in the same paragraph that they are introduced with the standard `where f_w is a neural network with weights w` style. Regarding explaining x and theta, we generalized the examples in the first paragraph of the introduction by modifying the sentence such that it reads `This problem setting occurs across scientific domains \cite{cole2021fast, alsing2018massive, brehmer2018constraining, hermans2020towards, lensing} where $\btheta$ generally represents input parameters of the simulator and $\bx$ the simulated output observation.`

---

> > ### Author Response · Authors · 2022-07-30
> > **initial reply to gFMN - 02**
> >
> > ### Questions
> >
> > -   We want to thank you for the question about limitations of NRE-C which are always important to clarify. We will break down our response down by point. Before that, one important theme in your question is about the efficiency of NRE-C compared to NRE-B regarding the generation of training data. In all training regimes presented in this paper, NRE-C and NRE-B use the same methods for simulating data. The fundamental difference lies in which training data is shown to the classifier and how the loss is computed.
> >
> >     Since this is a natural question, we added a paragraph to Appendix A where we discuss how to sample from the distributions. This will hopefully increase the clarity of the paper and contribute to its self containment.
> >
> >
> > -   The introduction of the independently drawn class introduces only negligible computational complexity. You raise a good question about how one samples from p(x), noting that is may be difficult. This is a natural conclusion because computing the value of p(x) is incredibly expensive. However, sampling is, luckily, not difficult. Consider two pairs (t0, x0) and (t1, x1) both sampled from p(t, x). If the t’s are swapped then the sample looks like (t1, x0) and (t0, x1) and the new pairs are sampled from p(t)p(x). This is exactly what we mean when we say “bootstrap” to generate the samples we need in the loss function. It is also a standard technique used by both NRE-A and NRE-B. For the price of reordering an array, we can sample from the independent distribution!
> >
> >     We added an entire paragraph about this in Appendix A to clarify. It’s called `Sampling in \SBI`. It will be uploaded with the new PDF.
> >
> > -   I’m not sure where we said that sampling from the prior is slow, would you mind pointing it out? What you read is most likely a typo and we’d be happy to correct it! It is often the case that drawing from the prior is trivial and simulating the data x can be very expensive. We note this in the first sentence of Section 3.2 “Leveraging fast priors (drawing theta).”
> >
> > -   This is a good question to check the efficiency of NRE-B and NRE-C. If the efficiency is defined as `posterior accuracy / simulations` then NRE-B is less efficient with the right hyperparameters. The construction of this extra independently drawn class does not introduce any new simulations since the budget is (in the standard case and Section 3.3 “Simulation-based inference benchmark”) fixed beforehand and the independently drawn samples are bootstrapped. The fundamental difference is in the loss functions. If the efficiency was computed as `posterior accuracy / total computation` then we’d note that computing the loss for NRE-C is negligibly slower than NRE-B; however, this is likely on the order of nano or milliseconds per epoch.
> >
> >
> >
> >
> > ### Limitations
> >
> > -   We will make the broader impact statement less broad by giving a few examples. We changed the last sentence to `This nuance can be missed by practitioners doing inference in any field;  however, special care should be taken when producing inference results that may be used for making decisions in areas like predicting hidden variables that describe human behavior or determining what factors are responsible for climate change. This list is non-comprehensive and not specific to \SBI.`
> >
> >
> >
> >
> > ### Conclusion
> >
> > Thank you again for your insight and commitment to clarity and improving the quality of our work. We hope that by addressing your concerns about self containment and clarifying the figures and notation you will consider raising your score to reflect the improvements that we’ve made. Furthermore, we hope that we adequately addressed your questions. We’d be happy to comment further if there are remaining points!

---

### Author Response · Authors · 2022-07-30
**general reply**

We want to thank all three reviewers for sharing their expertise and impressions of our work along with taking the time to read our paper. The positive impressions of the reviewers heartened us. We believe that the suggestions that were made in the reviews were thoughtful and relevant, thereby improving the paper further.

We took action by changing the text as suggested by the reviewers. Since the requests were primarily to include more information / clarification or to move information from the supplemental material into the main text, we moved the broader impact section to Appendix A to free up space. Our revised paper still fits within the required 9 page limit and the pdf has been updated (along with an updated appendix).

The only matters raised by reviewers that we have not finished our actions on yet are:
The averaging of the C2ST over both data and seed suggested by reviewer 1x3V. The reason is that the additional experiments are still running. We expect that the new results will not significantly change our reported C2ST as we have already averaged over other aspects, namely across posteriors (data) and across tasks (in the main c2st result table).
Transferring details about other paper’s benchmark experiments into our main text. Assuming both acceptance and the allowance of an additional content page, then we will address this matter as best we can given the space provided.
The first matter is still open. We will update the results when the experiments finish.

In the meantime, we look forward to a fruitful discussion of the paper.

---

### Author Response · Authors · 2022-08-04
**new results**

Dear reviewers,

We present here a new version of table 1 where the average has been taken over both seed (with four initializations), various posteriors (data), and across tasks. This was requested by reviewer 1x3V.

| Simulation budget | $10^3$ | $10^4$ | $10^5$ |
|-------------------|--------|--------|--------|
| Algorithm         |        |        |        |
| \NREC (ours)      | 0.856  | 0.760  | 0.682  |
| \REJABC           | 0.965  | 0.920  | 0.871  |
| \NLE              | 0.826  | 0.753  | 0.723  |
| \NPE              | 0.838  | 0.736  | 0.654  |
| \NRE (\NREB)      | 0.867  | 0.811  | 0.762  |
| \SMCABC           | 0.948  | 0.873  | 0.816  |
| \SNLE             | 0.783  | 0.704  | 0.655  |
| \SNPE             | 0.796  | 0.677  | 0.615  |
| \SNRE (\SNREB)    | 0.788  | 0.703  | 0.610  |

We are happy to present a version of table 3 with the result for each task individually, but it is quite long. If any reviewer would like this, we'd be happy to provide the table.

---

### Meta-Review · Area_Chair_TpE9 · 2022-08-24

**Recommendation:** Accept
**Confidence:** Certain

**Metareview:**

The three reviewers agreed that the work is a valuable contribution to its field, and presents extensive experiments.

For the readers' benefit, I kindly ask the authors to take into account reviewers comments while preparing the camera-ready version.
In particular, the revised version should include:
   - the updated results table (across seeds, per dataset);
   - a clearer formatting of Figure 5;
   - an expanded discussion points on (i) how their method compares against learning the bias of NRE-B (Ma and Collins, 2018) (ii) clarifying the part on sequential vs. amortized methods.

**Award:**

No

---

### Decision · Program_Chairs · 2022-09-14

Accept